

# An ensemble Kalman filter data assimilation system for the whole neutral atmosphere

Dai Koshin[1], Kaoru Sato[1], Kazuyuki Miyazaki[2, 3], Shingo Watanabe[3]

[1]Department of Earth Planetary Science, The University of Tokyo, Tokyo, Japan
[2]Jet Propulsion Laboratory, California Institute of Technology, Pasadena, CA, USA
[3]Japan Agency for Marine-Earth Science and Technology, Yokohama, Japan

*Correspondence to*: Dai Koshin (koshin@eps.s.u-tokyo.ac.jp)

**Abstract.** A data assimilation system with a four-dimensional local ensemble transform Kalman filter (4D-LETKF) is developed to make a new analysis data for the atmosphere up to the lower thermosphere using the Japanese Atmospherics
General Circulation model for Upper Atmosphere Research. The time period from 10 January 2017 to 20 February 2017, when an international radar network observation campaign was performed, is focused on. The model resolution is T42L124 which can resolve phenomena at synoptic and larger scales. A conventional observation dataset provided by National Centers for Environmental Prediction, PREPBUFR, and satellite temperature data from the Aura Microwave Limb Sounder (MLS) for the stratosphere and mesosphere are assimilated. First, the performance of the forecast model is improved by
modifying the vertical profile of the horizontal diffusion coefficient and modifying the source intensity in the non-orographic gravity wave parameterization, by comparing it with radar wind observations in the mesosphere. Second, the MLS observational bias is estimated as a function of the month and latitude and removed before the data assimilation. Third, data assimilation parameters, such as the degree of gross error check, localization length, inflation factor, and assimilation window are optimized based on a series of sensitivity tests. The effect of increasing the ensemble member size is also
examined. The obtained global data are evaluated by comparison with the Modern-Era Retrospective analysis for Research and Applications version 2 (MERRA-2) reanalysis data covering pressure levels up to 0.1 hPa and by the radar mesospheric observations which are not assimilated.

## 1 Introduction

It is well known that the earth's climate is remotely coupled: For example, when El Niño occurs, convective activity in the
tropics strongly affects mid-latitude climate with the appearance of the Pacific-North American pattern (Horel and Wallace, 1981). Convective activity in maritime continents also modulates mid-latitude climates by generating the Pacific-Japan pattern (Nitta, 1987). Most of these climate couplings between the tropics and mid-latitude regions are caused by the horizontal propagation of stationary Rossby waves (Holton and Hakim, 2013). Teleconnection through stratospheric processes has also been known. For example, the sea-level pressure in the Arctic rises during El Niño. It was shown that this
teleconnection occurs by modulation of planetary wave intensity and propagation in the stratosphere (Cagnazzo and Manzini,



2009). It is also well known that the occurrence frequency of stratospheric sudden warming (SSW), which exerts a strong influence on the Arctic oscillation of sea-level pressure (Baldwin and Dunkerton, 2001), is high during the easterly phase of the quasi-biennial oscillation in the equatorial stratosphere (Holton and Tan, 1980). This is also due to the modulation of the propagation of planetary-waves in the stratosphere. Thus, the stratosphere is an important area that brings about the remote

coupling of climate.

Recently, the presence of interhemispheric coupling through the mesosphere has been reported as well. When the temperature in the polar winter stratosphere is high, the temperature in the polar summer upper mesosphere is also high with a slight delay (Karlsson et al., 2009). This coupling is clear for at least one-month average (Gumbel and Karlsson, 2011). The interhemispheric coupling, which is initiated by SSW in the winter hemisphere, occurs at shorter time scales (Körnich

and Becker, 2010). When the SSW occurs in association with the breaking of strong planetary waves originating from the troposphere, the westerly wind of the polar night jet significantly weakens or, in strong cases, even turns easterly. The critical level filtering of the gravity waves toward the mesosphere is then modulated, and the gravity-wave forcing that drives the mesospheric meridional circulation with an upward (downward) branch on the equatorial (polar) side becomes weak. Thus, the temperature in the equatorial region increases and the poleward temperature gradient in the summer

hemisphere weakens. The weak wind layer above the easterly jet in the summer hemisphere lowers so as to satisfy the thermal wind relation. The eastward gravity-wave forcing region near the weak wind layer also descends and the upward branch of the meridional circulation, which maintains extremely low temperature in the summer polar upper mesosphere, weakens.

However, there are few observational evidences of gravity-wave modulation in the mesosphere. The

Interhemispheric Coupling Study by Observations and Modeling (ICSOM: http://pansy.eps.s.u-tokyo.ac.jp/icsom/) is a project to understand mesospheric gravity-wave modulation associated with SSWs on a global scale through a comprehensive international observation campaign with a network of mesosphere-stratosphere-troposphere (MST), meteor, and medium frequency (MF) radars as well as complementary optical and satellite-borne instruments. Since 2016, four campaigns have been successfully performed.

In the ICSOM project, we are simultaneously proceeding a model study using a gravity-wave permitting high-top general atmospheric circulation model (GCM), that covers the entire troposphere and middle atmosphere (up to the lower thermosphere). However, this is not easy because the GCMs including the entire middle atmosphere are not yet sufficiently mature even for relatively low resolutions that do not allow explicit gravity-wave simulation (e.g., Smith et al., 2017). Therefore, verification of the GCMs by high-resolution observations is necessary. In the ICSOM project, by validating the

high-top GCM using data from the comprehensive international radar observation campaigns, it is expected to reproduce high-resolution global data with high reliability. Using this global data, we plan to confirm regional representation of gravity wave characteristics detected by each radar and deepen the understanding of interhemispheric coupling quantitatively with a resolution of gravity-wave scales.





Gravity-wave simulation research using high-resolution GCMs has been performed in the past (e.g., Hamilton et al.,
1999; Sato et al., 1999, 2009, 2012; Watanabe et al., 2008; Holt et al., 2016). However, reproducing gravity-wave fields in
the global atmosphere at a specific date and time requires significant effort (Eckermann et al., 2018; Becker et al., 2004).
Data assimilation up to the scale of gravity waves is ideal to create global high-resolution grid data sequentially. However,
current data-assimilation schemes work well for geostrophic motions such as Rossby waves but not necessary for
ageostrophic motions such as gravity waves. Recent studies (Jewtoukoff, et al., 2015; Ehard et al., 2018) reported that
gravity waves observed in the European Center for Medium-Range Weather Forecast (ECMWF) operational data are partly
realistic in the lower and middle stratosphere, but more validation with observation data is necessary. It has also been shown
that the difference in horizontal winds between reanalysis datasets is quite large in the equatorial region where the Coriolis
parameter becomes zero (Kawatani et al., 2016). The reasons for this problem may be the insufficient maturity of the models
to accurately express ageostrophic motions and/or the shortage of observation data including gravity waves to be assimilated.

Data assimilation for the mesosphere is particularly not easy partly because the energy ratio of Rossby waves and
gravity waves is reversed there (Shepherd et al., 2000) and partly because observational data for the mesosphere are
significantly limited compared to those for the lower atmosphere. In addition, it has been shown that, in the upper
stratosphere and the mesosphere, Rossby waves are generated in situ due to baroclinic/barotropic instability caused by wave
forcing associated with breaking or critical-level absorption of gravity waves propagating from the troposphere (Watanabe et
al., 2009; Ern et al., 2013; Sato and Nomoto, 2015; Sato et al., 2018). It has been found that gravity waves are
spontaneously generated in the middle atmosphere from the imbalance of the polar night jet (Sato and Yoshiki 2008; Snyder
et al., 2007; Shibuya et al., 2017), from an imbalance caused by the wave forcing due to primary gravity waves (Vadas and
Becker, 2018; Hayashi and Sato, 2018) and also by shear instability caused by primary gravity-wave forcing (Yasui et al.,
2018). The Rossby wave generation in the middle atmosphere due to primary gravity-wave forcing is regarded as a
compensation problem, which makes it difficult to understand the change in the Brewer-Dobson circulation in terms of the
relative roles of Rossby waves and gravity waves for climate projection with the models (Cohen et al., 2013). However,
these instabilities and the in-situ generation of waves in the middle atmosphere could significantly affect the momentum and
energy budget in the middle atmosphere and above (Sato et al., 2018; Becker, 2017). Hence, it is necessary to understand the
roles of these waves as accurately as possible based on credible, high-resolution model simulations validated by high-
resolution observations.

In view of the situation described above, the following method may be one of the best existing ways to create high-
resolution data of the entire middle atmosphere including gravity waves, for understanding teleconnection through the
mesosphere. First, a data assimilation is performed using a high-top but relatively low-resolution model to create grid data of
the real atmosphere from the ground to the lower thermosphere including only larger-scale phenomena such as Rossby
waves. Second, the analysis data obtained by the assimilation are used as initial values for a free run of high-resolution
GCMs to simulate gravity waves. Eckermann et al. (2018) and Becker and Vadas (2018) have performed pioneering studies
on the effectiveness of such free runs.



Reanalysis data over a long time period are produced using modern data assimilation schemes and released by meteorological organizations for climate analysis. These include the ECMWF interim reanalysis (ERA-Interim; Dee et al., 2011) and the fifth reanalysis (ERA5; Hersbach et al., 2018) produced by a four-dimensional (4D)-variational assimilation scheme (Var); MERRA (Rienecker et al., 2011) and the following version 2 (MERRA-2; Gelaro et al., 2017) by the National Aeronautics and Space Administration (NASA) by a three-dimensional (3D)-Var; the National Centers for Environmental Prediction (NCEP) Climate Forecast System Reanalysis (CFSR; Saha et al., 2010) and the Climate Forecast System version 2 (CFSv2; Saha et al., 2014); and the Japanese 55-year reanalysis (JRA-55; Kobayashi et al., 2015) by a 4D-Var. The ERA-Interim and JRA-55 cover up to a pressure of 0.1 hPa, the NCEP/CFSR and NCEP/CFSv2 up to 0.266 hPa, and MERRA, MERRA2 and ERA5 up to 0.1 hPa. However, global data of the middle and upper mesosphere to the lower thermosphere are not created regularly. As stated above, considering the importance of ageostrophic motions in the mesosphere and lower thermosphere (MLT), the data assimilation used for such meteorological organizations may not work very well for the middle stratosphere and above (Polavarapu et al., 2005). Therefore, in recent years, significant efforts have been made to assimilate data using GCMs which include the MLT region. Currently, the data available for studying the MLT region come from the Aura Microwave Limb Sounder (Aura MLS; beginning in 2004), Thermosphere Ionosphere Mesosphere Energetics and Dynamics (TIMED) Sounding of the Atmosphere using Broadband Emission Radiometry (SABER; beginning in 2002), and the Defense Meteorological Satellite Program (DMSP) Special Sensor Microwave Imager/Sounder (SSMIS; Swadley et al., 2008).

Global data of the atmosphere including the MLT region is valuable from the following viewpoints. First, it can improve prediction of the polar stratosphere (e.g., Hoppel et al., 2008; 2013, Polavarapu et al., 2005). It seems that anomalies in the MLT region start about one week earlier than stratospheric anomalies such as SSWs, propagating down to the troposphere. Thus, better understanding of the MLT physics and chemistry has a potential to improve long-range weather forecasts. Second, it is possible to quantitatively understand the transport of minor species from the MLT region (e.g., Hoppel et al., 2008; Polavarapu et al., 2005). For example, high-energy particles originating from the upper atmosphere contribute to the production of $NO_x$ which modulates the ozone chemistry in the stratosphere. Thus, the quantitative evaluation of the transport of such species is important for the prediction of the ozone layer. Third, it contributes to space-weather prediction, particularly for the prediction of the near-space environment (e.g., Hoppel. et al., 2013). Atmospheric waves excited in the lower and middle atmosphere, including gravity waves, Rossby waves, and tides, are main drivers of the general circulation in the height range of 100–150 km in the lower thermosphere (e.g., Akmaev, 2011; Miyoshi and Yigit, 2019). Thus, it is important to examine the properties of these waves in the mesosphere. Last but not least, it is interesting to understand middle atmosphere processes as a pure science (e.g., Hoppel et al., 2008).

The first attempt to create analysis data for the whole middle atmosphere using data assimilation was made by a Canadian group. They employed 3D-Var using the Canadian Middle Atmosphere Model (CMAM) with full interactive chemistry and nonlocal thermodynamic equilibrium (non-LTE) radiation (Polavarapu et al., 2005; Nezlin et al., 2009). The assimilation of the data in the troposphere and stratosphere has been shown to improve the analysis of large-scale



phenomena (zonal wavenumber $s < 10$) in the mesosphere (Nezlin et al., 2009). The daily mean time series from their data assimilation are validated by radar observations (Xu et al., 2011). Sankey et al. (2007) used the CMAM to carefully discuss the effectiveness of digital filters in the data assimilation. A series of studies at the Naval Research Laboratory (NRL) is

remarkable. Hoppel et al. (2008) performed the first mesospheric data assimilation at the Advanced Level Physics and High-Altitude (ALPHA) prototype of the Navy Operational Global Atmospheric Prediction System (NOGAPS) using a 3D-Var assimilation system (NAVDAS). After that, they introduced a 4D-Var to assimilate data using the NRL Navy Global Environmental Model (NAVGEM), a successor of NOGAPS (Hoppel et al., 2013). In this system, the SSMIS data was also assimilated along with the SABER and Aura MLS data. The calculation of the background error covariance matrix was

accelerated by introducing ensemble forecasts, and assimilation shocks to the model were reduced by using digital filters (McCormack et al., 2017; Eckermann et al., 2018). Global data with short time intervals were made by combining model forecasts with the assimilation products, and both short-term and annual variations of diurnal migrating tides were successfully captured (McCormack et al, 2017; Dhadley et al., 2018; Eckermann et al., 2018). These assimilation data products are utilized for the study of the quasi-two-day waves and five-day waves, as well as tides (Eckermann et al., 2009;

Pancheva et al., 2016; Eckermann et al., 2018), and for observation projects such as Deep Propagating Gravity Wave Experiment (DEEPWAVE; Fritts et al., 2016). A data assimilation study using the Whole Atmosphere Community Climate Model (WACCM) at the National Center for Atmospheric Research (NCAR) has been also conducted. Pedatella et al. (2014b) applied a Data Analysis Research Testbed (DART) Ensemble adjustment Kalman Filter (EAKF), which is a 3D-Var combined with a statistical scheme, to the WACCM and made analysis data for the largest recorded SSW event, which

occurred in 2009. They indicated that better analysis of the mesosphere requires assimilation of the mesospheric observational data. Similar discussion was made by Sassi et al. (2018) using the Specified Dynamics (SD)-WACCM, in which a nudging method was implemented. The reality of the analysis highly depends on the model's performance in the MLT region. One of the critical components to determine the MLT region in the model is gravity wave parameterizations (Pedatella et al., 2014a; Smith et al., 2017). According to Pedatella et al. (2018), the analysis of the SSW in 2009 by the

WACCM using DART showed that the expression of the downward transport of chemical components by the data assimilation is better than by the nudging method.

Nowadays whole-atmosphere models covering the surface to the exosphere have been developed (Akmaev, 2011). Data-assimilation or data-nudging studies using a whole-atmosphere model has been performed focusing on the SSW in 2009. These include studies using the whole atmosphere data assimilation system (WDAS), which includes the whole

atmosphere model and a 3D-Var analysis system (Wang et al., 2011), the Ground-to-topside model of Atmosphere and Ionosphere for Aeronomy (GAIA) with a nudging method (Jin et al., 2012), and SD-WACCM (Chandran et al., 2013; Sassi et al., 2013). Outputs from a long-term run using GAIA, which was nudged to the reanalysis data up to the lower stratosphere, were used for a momentum budget analysis in the whole middle atmosphere, and the importance of in-situ generation of gravity waves and Rossby waves in the middle atmosphere was suggested (Sato et al., 2018; Yasui et al., 2018).





Although most 4D data assimilation studies described above used 4D-Var, the method using an ensemble Kalman filter is also possible. The 4D-Var codes need to be developed for each model. In contrast, the 4-Dimensional Local Ensemble Transform Kalman Filter (4D-LETKF) developed by Miyoshi and Yamane (2007), which is a statistical assimilation method, is versatile and can thus be implemented in any model relatively easily. This study develops an assimilation system using the 4D-LETKF with a GCM with a top in the lower thermosphere. As the first step of the ICSOM

project, we used a low-resolution version of the GCM and examined the optimal parameters of the assimilation system for the middle atmosphere (i.e., the atmosphere up to the turbopause [~100 km]), as no studies employ the 4D-LETKF to assimilate data for such a high atmospheric region. The observation datasets used for the data assimilation are Aura MLS (v.4.2) temperature, which covers the whole stratosphere and mesosphere, and NCEP PREPBUFR, which is a standard dataset for the troposphere and lower stratosphere. The target time period is from January to February 2017, which includes

the second ICSOM observation campaign. On 1 February 2017, the criteria of the major SSW were satisfied. The structure of this paper is as follows. Section 2 describes the forecast model, observation data, and data assimilation system. Section 3 presents the results of the parameter assessment. Section 4 presents the results of analysis regarding fields in the middle atmosphere in ICSOM-2 using data from the best parameter setting. Section 5 gives the summary and concluding remarks.

## 2 Methodology

### 2.1 Forecast Model

We used the Japanese Atmospheric GCM for Upper Atmosphere Research (Watanabe and Miyahara, 2009) as a forecast model, which we refer to as "JAGUAR" in this paper. This model has a high model top of approximately 150 km and is based on the T213L256 middle atmosphere GCM developed for the Kanto project (Watanabe et al., 2008) and the Kyushu-GCM (e.g., Yoshikawa and Miyahara, 2005). This model uses important physical parameterizations for the MLT region such

as radiative transfer processes, including non-LTE and solar-radiative heating due to molecular oxygen and ozone. The effects of ion-drag, chemical heating, dissipation heating, and molecular diffusion are also parameterized in the model. In this study, a standard-resolution JAGUAR with a triangularly truncated spectral resolution of T42 corresponding to a horizontal resolution of about 300 km (a latitudinal interval of 2.8125 degrees) is used for the assimilation. The model has 124 vertical layers with a uniform vertical spacing of approximately 1 km in the middle atmosphere and 100–800 m in the

troposphere (see Figure A1 of Watanabe et al., 2015 for the vertical layers). Unlike a high-resolution JAGUAR, which resolves a certain portion of gravity waves (Watanabe and Miyahara, 2009), gravity waves are sub-grid scale phenomena for a standard-resolution JAGUAR. For this reason, both orographic (McFarlane, 1987) and non-orographic (Hines, 1997) gravity wave parameterizations are used. The wave-source distribution of non-orographic parameterization is given based on the results of a gravity-wave-resolving high-resolution GCM (Watanabe, 2008), and the intensity of the source is treated as

one of the tuning parameters. Horizontal diffusion is set as an e-folding time of 0.9 days for the minimum resolved wave length in the troposphere and stratosphere, and it exponentially increases with increasing height over the MLT region. In this



study, the vertical profile of horizontal diffusion above the stratopause is also treated as one of the tuning parameters. The monthly ozone mixing ratio from United Kingdom Universities Global Atmospheric Modeling Programme (UGAMP; Li and Shine, 1999) and monthly sea surface temperature and sea ice concentration from Met Office Hadley Centre's sea ice and sea

surface temperature data set (HadISST; Rayner et al., 2003) are linearly interpolated in time and used as boundary conditions.

## 2.2 Measurements used in the assimilation

### 2.2.1 PREPBUFR

One of the observation data used for the assimilation is the PREPBUFR global observation dataset compiled by the National

Centers for Environmental Prediction and archived at the University Corporation for Atmospheric Research (https://rda.ucar.edu/datasets/ds337.0/) which includes surface pressure as a function of longitude and latitude, and temperature, wind, and humidity as functions of longitude, latitude, and pressure (or height) from radiosondes, aircrafts, wind profilers, and satellites. Ground-based observations are mainly distributed in the height range from the ground to the lower stratosphere, and approximately 70% of the data are taken at stations located in the Northern Hemisphere. Since May

1997, daily data has been uploaded with a delay of several days. The number of data per one assimilation step (every 6 h) is 1,000-20,000 for balloon-borne radiosonde measurements, ~1,000 for aircraft measurements, ~40,000 for satellite wind measurements, ~10,000 for meteorological radar measurements, ~50,000 for measurements at the ground, and ~500,000 for sea scatterometer measurements.

The observation errors provided in the PREPBUFR dataset as a function of the type of measurements and altitude[1]

were used in the data assimilation. For example, the observation errors in radiosonde temperature data are 1.2 K at 1000 hPa, 0.8 K at 100 hPa, and 1.5 K at 10 hPa. The horizontal resolution of the GCM used in this study is not sufficient to represent the fine structure captured by these observations. Representativeness errors, which come from the difference in resolutions between individual measurements and the model, could degrade the data assimilation performance. If representation errors are random and large numbers of observations are assimilated, their impact could be negligible. Because substantial numbers

of observations are available within a model grid cell in a data assimilation cycle in our analysis, the observation data were thinned before assimilation to reduce the computational cost of the data assimilation analysis. One-fourth of data from aircraft measurements and satellite winds, and radiosonde data at the standard pressure levels of 1000, 925, 850, 700, 500, 400, 300, 250, 200, 150, 100, 70, 50, and 10 hPa were used for the data assimilation.

---

[1] http://www.emc.ncep.noaa.gov/mmb/data_processing/obserror.htm





### 2.2.2 Aura MLS

225 The MLS instrument onboard the Aura satellite was launched in 2004. The satellite takes the polar orbit 14 times a day. Vertical profiles of several atmospheric parameters are retrieved from a limb sounding of the thermal emissions of the atmosphere. We used temperature data (v.4.2) retrieved from the radiation of oxygen ($O_2$; 118GHz) and oxygen isotope $O^{18}O$; 239GHz) of Aura MLS (Livesey et al., 2018) for the assimilation. The data are distributed at 55 vertical layers from 261 to 0.001 hPa at ~ 2 km intervals. The estimated retrieval errors are ~ 0.5 K at 261–10 hPa, ~ 1 K at 10–0.3 hPa, ~ 2 K

230 for 0.3–0.04 hPa, and ~ 3 K for 0.04–0.001 hPa. For the observation operator, we included weighting functions (called "averaging kernels") to consider the vertical sensitivity of the measurements. The weighting functions at the equator and at 70˚N are available on the Aura MLS mission website (https://mls.jpl.nasa.gov/data/ak/). Assuming that the measurement vertical sensitivity is invariant for a wide area, the averaging kernel for the equator and that for 70˚N are respectively applied to the latitudinal range of 40˚S–40˚N and the remaining high latitude regions (i.e., 40˚N–90˚N and 40˚S–90˚S).

235  The horizontal intervals of the Aura MLS observation data along the track, which is almost parallel to the meridional direction, are approximately 2˚, and so two or three profiles are included in the area represented by a grid point of the forecast model. To reduce the computational cost of the data assimilation, the observations are horizontally averaged to reduce the resolution comparable to the forecast model resolution before the assimilation, without considering any correlation between individual observation errors. Errors in the retrievals in some parameters can be correlated in space, but

240 their quantitative estimates are difficult. The measurement error is used as the diagonal component of the observation error covariance matrix.

  It has been suggested that the Aura MLS data includes observation bias (e.g., Randel et al., 2016). In this study, a bias correction is performed and the effect of the bias correction on the analysis data is examined. In addition to the retrieval quality flag information, a gross error check was applied in the quality control to exclude observations that are far from the

245 first guess. The optimal settings for the gross error check are considered to be different between the mesosphere and lower atmosphere because of the different growth rates of model error in a specific period of time (e.g., a data assimilation window). Thus, the appropriate degrees of the gross error check are also examined.

### 2.3 Data assimilation system

 The 4D-LETKF (Miyoshi and Yamane, 2007) is used as a data assimilation method. This method is an extension of the 3D-

250 LETKF (Hunt et al., 2007) which includes the dimension of time (4-D ensemble Kalman filter; Hunt et al., 2004). The base of the program used in this study has already been applied to many types of forecast models, such as the Global Spectral Model (GSM; Miyoshi and Sato, 2007), the Atmospheric GCM for Earth Simulator (AFES; Miyoshi et al., 2007), and the Non-hydrostatic Icosahedral Atmospheric Model (NICAM; Terasaki et al., 2015).

  This section introduces the formulas used in the 4D-LETKF. The analyses, forecasts, and observations are denoted

255 by $\mathbf{x}^a$, $\mathbf{x}^f$, and $\mathbf{y}^o$, respectively. The optimal value of $\mathbf{x}^a$ is derived from $\mathbf{x}^f$ and $\mathbf{y}^o$ by the following equation:


$$\mathbf{x}^a = \mathbf{x}^f + \mathbf{K}(\mathbf{y}^o - \mathbf{H}\mathbf{x}^f) = \mathbf{x}^f + \mathbf{K}\mathbf{d}, \tag{1}$$

where, $\mathbf{K}$ is a weighting function, $\mathbf{d}$ ($\equiv \mathbf{y}^o - \mathbf{H}\mathbf{x}^f$) is the innovation, and $\mathbf{H}$ is an observation operator that converts the model space variables into observational space variables. For assimilating MLS retrievals, the observation operator includes the averaging kernel and the spatial operator. The second term on the right-hand side represents data assimilation corrections

(i.e., increments). Using the differences from the true value ($\mathbf{x}^t$), $\delta\mathbf{x}^a = \mathbf{x}^a - \mathbf{x}^t$, $\delta\mathbf{x}^f = \mathbf{x}^f - \mathbf{x}^t$, and $\delta\mathbf{y}^o = \mathbf{y}^o - \mathbf{H}\mathbf{x}^t$, so Eq. (1) can be rewritten as follows:

$$\delta\mathbf{x}^a = \delta\mathbf{x}^f + \mathbf{K}(\delta\mathbf{y}^o - \mathbf{H}\delta\mathbf{x}^f) = (\mathbf{I} - \mathbf{K}\mathbf{H})\delta\mathbf{x}^f + \mathbf{K}\delta\mathbf{y}^o, \tag{2}$$

where $\mathbf{I}$ is an identity matrix. The analysis error covariance is defined as

$$\mathbf{P}^a \equiv \langle \delta\mathbf{x}^a (\delta\mathbf{x}^a)^T \rangle = (\mathbf{I} - \mathbf{K}\mathbf{H})\mathbf{P}^f(\mathbf{I} - \mathbf{K}\mathbf{H})^T + \mathbf{K}\mathbf{R}\mathbf{K}^T, \tag{3}$$

where $\mathbf{P}^f \equiv \langle \delta\mathbf{x}^f (\delta\mathbf{x}^f)^T \rangle$ is the forecast error covariance and $\mathbf{R} \equiv \langle \delta\mathbf{y}^o (\delta\mathbf{y}^o)^T \rangle$ is the observation error covariance. The correlation between the forecast error and the observation error is supposed to be zero ($\langle \delta\mathbf{x}^f (\delta\mathbf{y}^o)^T \rangle = 0$). The optimal $\mathbf{x}^a$ should minimize the summation of the analysis error covariance ($\mathrm{tr}(\mathbf{P}^a)$). This means that

$$\frac{\partial}{\partial \mathbf{K}}\mathrm{tr}(\mathbf{P}^a) = 0. \tag{4}$$

Solving Eq. (4) with respect to the weight matrix $\mathbf{K}$ yields

$$\mathbf{K} = \mathbf{P}^f\mathbf{H}^T(\mathbf{H}\mathbf{P}^f\mathbf{H}^T + \mathbf{R})^{-1} \tag{5}$$

and the analysis $\mathbf{x}^a$ is derived by Eq. (1). The weight matrix $\mathbf{K}$ is called the "Kalman gain". Using $\mathbf{K}$, the analysis error covariance $\mathbf{P}^a$ is rewritten as

$$\mathbf{P}^a = (\mathbf{I} - \mathbf{K}\mathbf{H})\mathbf{P}^f, \tag{6}$$

which gives the relationship $\mathbf{K} = \mathbf{P}^a\mathbf{H}^T\mathbf{R}^{-1}$.

The size of $\mathbf{P}^f$ and $\mathbf{P}^a$ is the square of the degree of freedom in the model. Thus, for systems with huge degrees of freedom, such as GCMs, the calculation of $\mathbf{P}^f$ and $\mathbf{P}^a$ requires a large computational cost. This problem is avoided by replacing the forecast, analysis and each error with the mean and variance for $m$ members of an ensemble. This is called the "Ensemble Kalman Filter (EnKF; Evensen, 2003). The ensemble mean $\bar{\mathbf{x}}$ and background error covariance matrix $\mathbf{P}$ are written as follows:

$$\bar{\mathbf{x}} = \frac{1}{m}\sum_{i=1}^{m}\mathbf{x}_i, \tag{7}$$

$$\mathbf{P} = \langle \delta\mathbf{x}(\delta\mathbf{x})^T \rangle \approx \frac{1}{m-1}\sum_{i=1}^{m}(\mathbf{x}_i - \bar{\mathbf{x}})(\mathbf{x}_i - \bar{\mathbf{x}})^T = \frac{1}{m-1}\sum_{i=1}^{m}\delta\mathbf{x}_i(\delta\mathbf{x}_i)^T. \tag{8}$$

However, with a limited number of ensembles, the forecast error tends to be underestimated in a system with a large degree of freedom. A variety of methods have been proposed to overcome this problem (e.g., Whitaker et al., 2008). In our study, the forecast ensemble perturbation ($\delta\mathbf{x}^f$) is multiplied by the factor $F$, which is a little larger than 1 ($F = 1 + \Delta$; $\Delta$ is called

an "inflation factor").





$$\delta \mathbf{x}_i^f \leftarrow (1 + \Delta)\, \delta \mathbf{x}_i^f \tag{9}$$

is employed and the $\Delta$ value is optimized.

The Kalman gain is simply written by using $\mathbf{E}$, which is the root of $\mathbf{P}$. Using

$$\sqrt{m-1}\,\mathbf{E} \equiv [\delta\mathbf{x}_1 \mid \dots \mid \delta\mathbf{x}_m], \tag{10}$$


$$\mathbf{K} = \mathbf{P}^f\mathbf{H}^{\mathrm{T}}(\mathbf{H}\mathbf{P}^f\mathbf{H}^{\mathrm{T}} + \mathbf{R})^{-1} = \mathbf{E}^f(\mathbf{H}\mathbf{E}^f)^{\mathrm{T}}[\mathbf{H}\mathbf{E}^f(\mathbf{H}\mathbf{E}^f)^{\mathrm{T}} + (m-1)\mathbf{R}]^{-1} \tag{11}$$

is derived. Further manipulation yields another expression of $\mathbf{K}$:

$$\mathbf{K} = \mathbf{E}^f[(m-1)\,\mathbf{I} + (\mathbf{H}\mathbf{E}^f)^{\mathrm{T}}\mathbf{R}^{-1}\mathbf{H}\mathbf{E}^f]^{-1}(\mathbf{H}\mathbf{E}^f)^{\mathrm{T}}\mathbf{R}^{-1}. \tag{12}$$

To reduce the calculation cost, the inverse matrix of Eq. (11) or eq. (12) with a smaller size is chosen. Usually, as the number of the ensemble is much smaller than the number of observations, Eq (14) is used.

The LETKF treats the analysis error covariance matrix,

$$\widetilde{\mathbf{P}}^a \equiv [(m-1)\,\mathbf{I} + (\mathbf{H}\mathbf{E}^f)^{\mathrm{T}}\mathbf{R}^{-1}\mathbf{H}\mathbf{E}^f]^{-1} \tag{13}$$

in the ensemble space. The relationship between this matrix in the ensemble space and the analysis error covariance matrix in the model space is expressed as $\mathbf{P}^a = \mathbf{E}^f\widetilde{\mathbf{P}}^a(\mathbf{E}^f)^{\mathrm{T}}$, and $\mathbf{E}^a = \mathbf{E}^f(\widetilde{\mathbf{P}}^a)^{\frac{1}{2}}$ is the ensemble update. In this way, the analysis $\mathbf{x}_i^a$ is obtained as


$$\mathbf{x}_i^a = \bar{\mathbf{x}}^a + \delta\mathbf{x}_i^a = \bar{\mathbf{x}}^f + \delta\mathbf{x}_i^a + \mathbf{E}^f\widetilde{\mathbf{P}}^a(\mathbf{H}\mathbf{E}^f)^{\mathrm{T}}\mathbf{R}^{-1}\mathbf{d}\,. \tag{14}$$

Using the shape of the $N \times m$ matrix, where $N$ is the number of the variables, Eq. (14) is written as

$$[\mathbf{x}_1^a \mid \cdots \mid \mathbf{x}_m^a] = [\bar{\mathbf{x}}^f \mid \cdots \mid \bar{\mathbf{x}}^f] + \mathbf{E}^f\mathbf{W}, \tag{15}$$

where

$$\mathbf{W} = (\widetilde{\mathbf{P}}^a)^{\frac{1}{2}} + \left[\,\widetilde{\mathbf{P}}^a(\mathbf{H}\mathbf{E}^f)^{\mathrm{T}}\mathbf{R}^{-1}\mathbf{d} \mid \cdots \mid \widetilde{\mathbf{P}}^a(\mathbf{H}\mathbf{E}^f)^{\mathrm{T}}\mathbf{R}^{-1}\mathbf{d}\right]. \tag{16}$$

To avoid unrealistic correction caused by remote observations with the use of a limited ensemble size, a weighting function based on the distance from the analysis point is multiplied by the observation error. This method is called "localization". The calculation is independently performed at each other grid so it can be performed in parallel with high computational efficiency. The length of localization is also a setting parameter.

When an analysis ensemble is derived, each ensemble takes its own time evolution calculated by the forecast model,
and the forecast at the next step is derived by

$$\mathbf{x}_{i,t+1}^f = M\big(\mathbf{x}_{i,t}^a\big). \tag{17}$$

In this way, the forecast and analysis steps are repeated through the data assimilation cycles.

Here we extend to a 4-D analysis. By the modification of the observation operator, the observation at any time ($j2$) can be assimilated as the information of time development from the target time ($j1$). One such assimilation is called the 4D-
EnKF (Hunt et al., 2004).

The forecast at the time step $j1$ is written as a weighted mean of forecast ensembles:

$$\mathbf{x}_{j1}^f = [\mathbf{x}_{1,j1}^f \mid \cdots \mid \mathbf{x}_{m,j1}^f]\,\mathbf{w} \equiv \mathbf{X}_{j1}^f\mathbf{w}. \tag{148}$$





The weighting matrix $\mathbf{w}$ is unknown but is calculated by the pseudo-inverse matrix:

$$\mathbf{w} = \left((\mathbf{X}_{j1}^f)^{\mathrm{T}}\mathbf{X}_{j1}^f\right)^{-1}(\mathbf{X}_{j1}^f)^{\mathrm{T}}\mathbf{x}_{j1}^f. \tag{159}$$

On the other hand, the (unknown) forecast at the time step $j2$ is also written as a weighted mean of forecast ensembles:

$$\mathbf{x}_{j2}^f = \mathbf{X}_{j2}^f\mathbf{w}. \tag{20}$$

Substituting Eq. (16) into this equation, the following formula is obtained:

$$\mathbf{x}_{j2}^f = \mathbf{X}_{j2}^f\mathbf{w} = \mathbf{X}_{j2}^f\left((\mathbf{X}_{j1}^f)^{\mathrm{T}}\mathbf{X}_{j1}^f\right)^{-1}(\mathbf{X}_{j1}^f)^{\mathrm{T}}\mathbf{x}_{j1}^f. \tag{21}$$

Finally, the modified observation operator to assimilate the observation at time step $j2$ to the forecast at time step $j1$ is
written as follows:

$$\mathbf{H}' = \mathbf{H}\mathbf{X}_{j2}^f\left((\mathbf{X}_{j1}^f)^{\mathrm{T}}\mathbf{X}_{j1}^f\right)^{-1}(\mathbf{X}_{j1}^f)^{\mathrm{T}}. \tag{22}$$

Appendix A explains that directly assimilating the observation at a certain time step by the modified observation operator is
the same as assimilating at the time of observation and then calculating the time evolution after the assimilation. Thus, this
method is regarded as a kind of 4D assimilation including the information of the time development. Another advantage of
this method is that future observations can be assimilated, as is similar to the Kalman smoother. In this study, this extended
LETKF with 4D assimilation is used. The time interval (called the "assimilation window") between the observations and the
analysis is one of the setting parameters.

The EnKF initial condition is obtained using the time-lagged method as follows. First, a six-month free-run is
performed from a climatological restart file for June 1. The results from the free run over ten days with a center of January 1
are used as the initial condition for each ensemble member on January 1. The analysis data for the first 10 days of the
assimilation are regarded as a spin-up, and, hence, are not used to examine the assimilation performance.

## 2.4 The method of parameter validation in the data assimilation system

As already mentioned, the parameter set of data assimilation usually made for the troposphere and stratosphere is not
necessarily appropriate for the analysis when the MLT region are included. This is because the dominant physical processes
and scales of motions could be different (e.g., Shepherd et al., 2000; Watanabe et al., 2008). This section describes the
parameters that should be optimized for the data assimilation system for the whole neutral atmosphere from the troposphere
up to the lower thermosphere. The relevance criteria of the data assimilation for each parameter are also described.

The parameters included in the data assimilation system are divided into two categories. The first category includes
two parameters describing the GCM: the horizontal diffusion coefficient and the factor of gravity wave source intensity in
the gravity wave parameterization. The second category includes five parameters related to the data assimilation: the degree
of gross error check, the localization length, the inflation factor, the length of assimilation window, and the number of



ensembles. The sensitivity of the performance of assimilation is tested by changing one parameter among the standard set of the parameters as shown in Table 1. Finally, the performance of the assimilation with the best set of parameters is confirmed.

The criteria used for the evaluation of the data assimilation for each parameter setting are observation minus forecast (OmF) and observation minus analysis (OmA) in the observational space. One more criterion for examining the quality of data assimilation is $\chi^2$, which was introduced by Ménard and Chang (2000):

$$\chi^2 = \mathrm{tr}(\mathbf{YY}^{\mathrm{T}}), \qquad \mathbf{Y} = \frac{1}{\sqrt{m}}\big(\mathbf{y} - H(\bar{\mathbf{x}}^b)\big)\big(\mathbf{HE}^b(\mathbf{HE}^b)^{\mathrm{T}} + \mathbf{R}\big)^{-\frac{1}{2}}.$$

The parameter $\chi^2$ describes the consistency between the innovation with the covariance matrices for the model forecast and
the observations. The $\chi^2$ values should be close to 1 if the background and observation errors are properly specified in the assimilation system. The $\chi^2$ values higher (lower) than 1 mean that the background or observation error has been underestimated (overestimated) against the innovation in the observational space.

## 3 Results

In this section, two types of parameter sensitivity experiments are performed. One is a parameter tuning of the forecast
model to reduce the systematic biases of the model in the MLT region. The other is an optimization of parameters related to the data assimilation module. Table 1 summarizes the experiments performed so as to obtain an optimal parameter set. The optimal parameter set is the one denoted by "Ctrl". The grounds for regarding this parameter set as optimal are described in detail in the following subsections.

### 3.1 Forecast model improvement

To reduce the model bias in the mesosphere, the vertical profile of the horizontal diffusion coefficient and the gravity wave source intensity in the non-orographic gravity wave parameterization are examined by comparing observations in the summertime Antarctic mesosphere. Here, the zonal wind observed by an MST radar called the PANSY radar in the Antarctic (Sato et al., 2014) is used as a reference of the mesospheric wind. Note that the temporal and longitudinal variation of the dynamical field is relatively small in January and February in the summertime Antarctic mesosphere.

### 3.1.1 Horizontal diffusion coefficient

The downscale energy cascade from resolved motions to unresolved turbulent motions is represented by numerical diffusion in most atmospheric models. A fourth-order horizontal diffusion scheme is used in the present version of the JAGUAR to prevent the accumulation of energy at the minimum wavelength. However, it is difficult to directly constrain the horizontal diffusion coefficient with observational data. In the present study, the horizontal diffusion coefficient is set to be constant up
to the lower mesosphere and then exponentially increase above to reproduce realistic temperature and wind structures. As the





horizontal diffusion in the model top is sufficiently strong to damp small-scale disturbances including (resolved) gravity waves, a sponge layer, which is usually included at the uppermost layers of GCMs, is not used in the model.

To optimize the tuning parameters of the forecast model, a series of free-run experiments with three different profiles of horizontal diffusion coefficients are performed. The impact of the difference in the horizontal diffusion coefficient is examined focusing on the zonal mean zonal wind field. All experiments are started with the same initial conditions, which are obtained from a free-run simulation with climatological external conditions (hereafter referred to as "the climatological simulation").

Figure 1 shows the three vertical profiles of the horizontal diffusion coefficient. The horizontal axis denotes the e-folding time at the highest resolved wavenumber (total wavenumber $n = 42$). Note that a smaller value on the horizontal axis means stronger horizontal diffusion. The standard diffusion profile of the JAGUAR is denoted by "A": the horizontal diffusion coefficient is constant below ~65 km (0.1 hPa), and rapidly increases above. We performed experiments using two other vertical profiles of the horizontal diffusion coefficient denoted by "B" and "C". The diffusion in the B and C profiles is stronger than A below ~60 km, but the increase in the diffusion for B is small compared to A. The diffusion for B is smaller than A above ~105 km.

A free run was performed using the model with each diffusion coefficient profile. The model fields at 00:00UTC 5 January of the climatological simulation were used for the same initial condition for the three free-run experiments. The results are examined for the model fields averaged over 40 days from 00:00 January 12 to 23:59 February 20 shown Figure 2.

It is also worth noting that the vertical axis in Figure 2 denotes the geometric altitude. The log-pressure height vertical coordinate commonly used in GCM studies is approximately 5 km higher than the geometric height in the upper mesosphere at high latitudes. This difference is taken into account using the model's geopotential height as the vertical coordinate for comparison with the radar wind data, which are obtained as a function of geometric height.

The zonal wind for the experiment with the A profile is more eastward above 87 km and more westward below 85 km, than observations. As a result, the vertical shear below 87.5 km is unrealistically strong. In contrast, the results of the experiments with the B and C diffusion profiles show similar profiles as the observations. The difference between the B and C experiments is observed in the vertical shear of zonal wind in the displayed upper mesosphere, which is large for B and small for C. The vertical shear is more realistic for B, although the wind magnitude itself is more realistic for C. We take B because this experiment has zero wind layer around 87 km, which is absent in the C experiment, as the zero-wind layer is an important feature observed in the upper mesosphere. We expect that the model with the B diffusion coefficient profile produces realistic vertical wind shear and, hence, potentially produces realistic wind fields including the zero-wind layer using the data assimilation.

Figure 2b shows the results of the data assimilation experiments with the B and C diffusion profiles for the time period of 12 January to 20 February 2017. An optimal set of the parameters except for the diffusion profiles in the data assimilation, which will be shown later in detail, was used for these experiments. The results from the B experiment are more realistic in the vertical shear and the location of the zero-wind layer than those of the C experiment, although the





difference is not large. Further comparison is performed for the latitude-height cross section of zonal mean zonal wind and Eliassen and Palm (E-P) flux (Figure 3) from the data assimilation with the B (left) and C (right) diffusion profiles. The meridional structures for the zonal mean zonal wind and the E-P flux in the stratosphere are similar below ~70 km. The difference is observed above. The zonal mean zonal wind and E-P flux are strongly damped above 70 km for C because of strong diffusion given there. This is probably unrealistic. The vertically fine structure is observed for the E-P flux in mid-

latitude and high latitude regions from 90–100 km, which is probably not real but due instead to numerical instability. From these results, we concluded that the optimal profile of the horizontal diffusion coefficient is B.

### 3.1.2 Gravity wave source intensity

The source intensity in the model's non-orographic gravity wave parameterization is tuned as well. The amplitude of upward propagating gravity waves increases with increasing altitude due to an exponential decrease of the atmospheric density. In

the upper mesosphere, breaking gravity waves cause strong forcing to the background winds, which maintains the weak wind layer near the mesopause (Fritts and Alexander, 2003). As the gravity waves, which affect the mesosphere most in the summer, are non-orographically generated, we tuned the source intensity of non-orographic gravity wave parameterization. It is expected that high source intensity lowers the wave breaking level and hence lowers the weak wind layer around the mesopause.

425        Figure 4a compares vertical profiles of the zonal mean zonal wind at 68.37°S averaged for the time period of 12 January to 20 February from free runs with different source intensities of the non-orographic gravity wave parameterization. The original source intensity is denoted by P1.0 and the modified intensities are 0.5 and 0.7 times the original source intensity, denoted by P0.5 and P0.7, respectively. The vertical profiles of the mean and the standard deviation of the PANSY radar observation for the time periods of 12 January to 20 February of 2016 to 2018 are shown for comparison. The initial

condition is the same as for the experiments with different horizontal diffusion coefficients.

        As we expected, the zonal mean zonal wind is weaker and the height of the zero-wind layer is lower for stronger source intensity. It seems that the zonal wind is weaker and the zero-wind layer is lower for P1.0 than those in the observations.

        Figure 4b shows the results of the data assimilation experiments with P0.5, P0.7, and P1.0 for the time period of 12

January to 20 February 2017. For these experiments, an optimal set of the parameters except for the source intensity was used in the data assimilation that will be shown later in detail. Although all the profiles are consistent with observations within the standard deviation range, the profile for P0.7 is the most similar to observations in terms of the magnitude and the height of the zero-wind layer. From these results, we determined that most optimal source intensity is 0.7 times the original one (i.e., P0.7).





## 3.2 Aura MLS bias correction


According to the Aura MLS data quality document (Livesey et al., 2018), the MLS temperature data has a bias compared to the SABER ones as a function of the pressure, which is -5 to +1 K for pressure levels of 1–0.1 hPa, -3 to 0 K for 0.1–0.01 hPa, and -10 K for 0.001 hPa. Thus, before performing the data assimilation, the MLS observation bias was removed as much as possible. Note that most previous studies of data assimilation did not make this bias correction. In this study, first,

the MLS observation bias is estimated as a function of the calendar day at each latitude and each pressure in a range of 177.838 hPa to 0.001 hPa. As the reference for the correction, we used the TIMED SABER temperature data (v. 2.0), which are considered to have little observation bias, at least in the altitude range from 85–100 km, as confirmed by Xu et al., (2006), who used data from the sodium lidar at Colorado State University, providing the absolute value of the temperature. Xu et al., (2006) attributed the disagreement below 85 km to high photon noise contaminating the lidar observations. Thus, we used

the SABER temperature data for the Aura MLS bias estimation in the height range of 10–100 km.

The observation view of SABER is altered every ~60 days by switching between northward (50° S–82°N) and southward (82°S–50°N) view modes. Thus, the latitudinal coverage of the SABER data is narrow compared to the MLS data. However, both datasets overlap for a long time period sufficient for statistical comparison between them.

The bias is estimated using data from 2005 to 2015. The original vertical resolution of MLS (~1–5 km) is coarser

than that of SABER (~0.5 km). First, the vertical mean of the SABER data corresponding to the vertical resolution at each of the 55 height levels of MLS data is obtained. Next, by linear interpolation, data at the grid of 25˚ (longitude) × 5˚ (latitude) are made for the MLS and SABER data. The time interval of the two datasets is 3 hours. Whereas the longitudinal variation of the bias is small —1 K at the most—, there is large annual variation in addition to the dependence on latitude and height. Thus, the MLS bias $T_{\mathrm{bias}}$ is obtained as a function of the latitude ($y$), height ($z$), and calendar day ($t$).

$$T_{\mathrm{bias}}(y,z,t) = T_{\mathrm{mean}}(y,z) + A(y,z)\cos\left(2\pi\frac{t}{365}\right) + B(y,z)\sin\left(2\pi\frac{t}{365}\right),$$

where $T_{\mathrm{mean}}$ represents the annual mean and the coefficients $A$ and $B$ are estimated by the least-squares method.

Figure 5 shows the estimated MLS bias as a function of calendar day and latitude at 10, 1, 0.1, and 0.01 hPa. The MLS bias shows a strong dependence on the altitude and latitude and has an annual cycle with amplitudes of 2 to 4 K. Figure 6 shows the vertical profiles of global mean Aura MLS bias along with the bias reported in the data quality document

(Liversey et al., 2018). For example, the global mean MLS biases estimated by the present study are -1.6±0.7 K at 56.2 hPa, 2.0±1.4 K at 1 hPa, and -3.8±1.1 K at 0.316 hPa. These are comparable to the biases described in the data quality document, which are -2 to 0K at 56.2 hPa, 0 to 5 K at 1 hPa, and -7 to 4 K at 0.316. This consistency indicates the validity of using the SABER data as a reference to estimate the MLS bias.

To evaluate the effect of the bias correction, data assimilation was performed using the MLS data with and without

bias correction. Figure 7 compares the latitude and pressure section of the zonal mean temperature and zonal wind between the two analyses. The difference in zonal mean temperature between the two (Figure 7c) resembles the corrected bias (Figure 7d). In contrast, the difference in the zonal mean zonal wind is not very large (Figure 7g). This is because the latitudinal





difference in the bias, which largely affects the zonal mean zonal wind through the thermal wind balance, is not large
compared to the vertical difference. In our study, the bias correction of the MLS data is made before the data assimilation, as
in standard assimilation-parameter setting.

### 3.3 Data assimilation setting optimization for 30 ensemble members

A series of sensitivity tests were performed to obtain the optimal values of each parameter in the data assimilation system
with 30 ensemble members. This number of members is practical for the data assimilation up to the lower thermosphere with
current supercomputer technology. The examined assimilation parameters are the gross error coefficient, localization length,
inflation coefficient, and assimilation window length. The optimal parameter set obtained by the sensitivity tests is denoted
by "Ctrl" in Table 1. This section discusses why the Ctrl parameter set is optimal by showing the assimilation performance
of the test in which one of the assimilation parameters is changed from the Ctrl parameter set. Section 3.3.5 gives a short
summary of the data assimilation setting optimization for 30 ensemble members.

### 3.3.1 Gross error coefficient

The gross error check is a method of quality control (QC) for the observation data used for the assimilation. In this method,
an observation is assimilated only if its OmF is smaller than expected, assuming that the forecast model provides a
reasonable representation of the true atmosphere. In many previous studies, observations are not assimilated when the OmF
exceeds 3–5 times the observational error for the troposphere and stratosphere. However, this criterion may not be suitable
for the mesosphere and lower thermosphere, in which the systematic bias and predictability of the model is likely higher and
lower, respectively (e.g., Pedatella et al., 2014a). Thus, the maximal allowable difference between the MLS observations and
model forecasts normalized by the observational error, which is called the "gross error coefficient", is set at 20 (hereafter
referred to as the "G20") for the MLS measurements as a control experiment of the present study, whereas it is set at 5 for
the PREPBUFR dataset as in previous studies. Consequently, this setting uses most of the MLS observations to correct the
model forecast. To investigate the effect of the enlarged gross error check coefficient, the result for the G20 is compared to
the experiment with the gross error coefficient of 5 also for the MLS measurement (G5). Note that the other parameters
beside the gross error coefficient are taken to be the same for the G20 (Ctrl) and G5 (see Table 1).

Figure 8 compares the histograms of the OmF (a gray curve) and OmA (a black curve) for the G20 (left) and G5
(right) experiments at 0.1 hPa. For both settings, the mean OmF values are slightly negative, whereas both the mean bias and
standard deviations of the OmA are smaller than those of the OmF. As expected, the OmF is more widely distributed for the
G20 than for the G5. This reflects the inclusion of more observations for the assimilation with the G20. Although the OmF
distribution for the G20 is close to the normal distribution, the distribution of for the G5 seems distorted, probably by an
overly strict selection of observations close to the model forecasts, which can be seen from the number of assimilated
observations, as indicated by the area of the histogram in Figure 8, which is only a half or a third the number for the G20.





Figure 9 shows vertical profiles of the mean OmF, OmA, and χ² (see section 2.4). Absolute values of the mean OmA are smaller than those of the mean OmF at almost all levels for both the G20 and G5 experiments. The absolute values of the mean OmF for the G20 are 1.5–2 times larger than those for the G5, implying that more observations that deviate largely from the forecasts are assimilated for the G20. It is worth noting that the absolute value of the mean OmF tends to increase with height, indicating that the forecast is less reliable in the upper stratosphere and mesosphere. The χ² values with

the G20 are larger than those with the G5 at all levels, reflecting a larger OmF for the G20. Generally speaking, such large χ² values with the G20 suggest that optimizing observation error and forecast spread is required. However, considering the current immature stage of the forecast model performance in the upper mesosphere and above, we dared to permit the large χ² values with the G20, as it allows us to use a large number of observations which are sparse in the upper stratosphere and mesosphere. In fact, the correlation between our analysis and observation is greatly improved for the G20 compared with the

G5 (shown later in Figure 14). It will be shown later that the χ² values are improved by taking a larger number of ensemble members.

### 3.3.2 Localization length

In the LETKF, the observation error is weighted with the shape of Gaussian function (observational localization) of the distance ($d$) between the location of the observation and the grid point:

$$R' = R \cdot \exp\left[\frac{d^2}{2L^2}\right],$$

where $R$ and $R'$ are the original and modified observation error covariances, respectively, and $L$ is a length scale which describes the distance to which the observation is effective in the assimilation. The parameter $L$ is called the "localization length", which is one of the key parameters that determine the LETKF performance. For a forecast model with a high degree of freedom, as in the present study, a small number of ensemble members may cause sampling errors in the forecast error

covariance at a long distance (e.g., Miyoshi et al., 2014). The localization is introduced to reduce such spurious correlations at long distances.

    A sensitivity test is performed by taking $L$ =300 km (L300), $L$ =600 km (L600, Ctrl), and $L$ =1000 km (L1000) without changing the other parameters (see Table 1). Table 2 shows the root mean square (RMS) of the temperature difference from the (bias-corrected) MLS observations for each experiment at 10 hPa, 1 hPa, 0.1 hPa, 0.01 hPa and 0.005

hPa in the stratosphere and mesosphere. A smaller RMS means that observations are better assimilated. The RMS is smallest for the L300 at lower levels of 10 hPa and 1 hPa, for the L600 at 0.1 hPa, and for the L1000 at upper levels of 0.01 hPa and 0.005 hPa, suggesting that optimal localization length depends on the height.

    Figure 10 shows the vertical profiles of the mean OmF, OmA, and χ² for the L300, L600 (Ctrl), and L1000. The magnitude of the mean OmF is largest for the L1000 below 0.3 hPa and for the L300 above between the three experiments.

The OmA values are distributed around zero for L600, whereas they tend to be negative for the L1000, particularly at lower





levels, and tend to be positive for the L300, particularly at upper levels. The $\chi^2$ values are smallest for the L300 and largest for the L1000.

Based on the results, we employed the L600 as the optimal $L$ to obtain better analysis for all levels from the stratosphere to the upper mesosphere. Our assimilation does not necessarily provide the best analysis data for a limited

height region, but it does assure that the analysis has a sufficient, nearly uniform quality for the whole middle atmosphere.

### 3.3.3 Inflation coefficient

To avoid possible underestimations in the forecast error covariances due to the small number of ensembles used in the assimilation, a covariance inflation technique is employed [see Eq. (9)]. The inflation coefficient is generally set to ~10% for the tropospheric system (e.g., Miyoshi et al., 2007; Miyoshi and Yamane, 2007; Hunt et al., 2007). We tested three different

inflation coefficients, namely, 7% (I7), 15% (I15, Ctrl), and 25% (I25). Note again that the sensitivity test was conducted by changing the inflation coefficient only (see Table 1).

Figure 11 shows meridional cross sections of the zonal mean ensemble spread of temperature for each assimilation run. The ensemble spread for the I7 is about 1 K at most in the mesosphere and lower thermosphere, which is smaller than the observation accuracy (1–3 K). In contrast, the ensemble spreads for the I15 and I25 are distributed in the range of the

observation accuracy. The necessity of larger inflation coefficient is likely due to the large diffusion coefficient in the upper mesosphere and lower thermosphere used in the forecast model (Figure 1). However, a larger inflation coefficient leads to an unrealistically thin vertical structure of ensemble spreads in the lower mesosphere, which is conspicuous for the I25 (Figure 11).

The vertical profiles of the mean OmF, OmA, and $\chi^2$ for the I7, I15, and I25 are shown in Figure 12. The absolute

value of the OmF and OmA is the smallest for the I15 at most altitudes. The $\chi^2$ values are also small for the I15 at most altitudes. Thus, we employed the optimal inflation coefficient of 15 % (i.e., I15).

Interestingly, the range of the optimal inflation coefficient also depends on the height from a viewpoint of $\chi^2$: The $\chi^2$ values for the I15 are similar to those for the I25 but smaller than the I5 above 0.2 hPa, whereas they are similar to those for the I7 but smaller than the I25 below 0.2 hPa.

### 3.3.4 Assimilation window length

The length of the assimilation window, which is a time duration of forecast and observation to be assimilated during one assimilation cycle, is also examined. When the assimilation window is set to 6 h, the forecast is first performed for $t$ =0–9 h, and next the analysis at $t = 6$ h is obtained by the assimilation using the forecasts and observations for the time period of $t$ =3–9 h in the assimilation. Note that this assimilation scheme uses future information to obtain the analysis at a certain

time. The obtained analysis is used as an initial condition for the next forecast for 9 h (i.e., $t = 6$–15 h). By repeating these





processes of forecast and assimilation, an analysis is obtained every 6 h. Thus, the length of assimilation window determines the length of the forecast run as well as the analysis interval.

A longer window has the advantage that more observations are assimilated at once, while taking the predicted physically balanced time evolution of dynamical fields into account. However, the longer forecast length may increase
model errors. Moreover, the 4D-EnKF assumes a linear time evolution of the dynamical field during the assimilation window length. Thus, variations with strong nonlinearity or with timescales shorter than the assimilation window length are not taken into account. We tested assimilation windows of 3 h (W3), 6 h (W6, Ctrl), and 12 h (W12) (see Table 1). The W12 (W3) assimilation was performed using the forecasts for $t = 6\text{–}18$ h ($t = 2\text{–}4$ h) out of the forecast run over 18 (4) hours and the corresponding observations.

Figure 13 shows the vertical profiles of the mean OmF, OmA, and $\chi^2$ for the three assimilations. The mean OmF and OmA values for the W6 and W12 are larger than for the W3, suggesting larger forecast errors in longer windows. This is probably because relatively short-period disturbances such as quasi-two day waves are dominant particularly in the mesosphere (e.g., Pancheva et al., 2016), which requires a short assimilation window for their expression. However, $\chi^2$ is significantly larger for the W3 than for the W6 and W12 particularly below 0.2 hPa, suggesting that the 3 h window may
have been too short to develop forecast error spreads sufficiently, especially for the lower atmosphere. Based on these results, the length of 6 h is regarded as the optimal assimilation window.

### 3.3.5 Comparison of a series of sensitivity tests for data assimilation with 30 ensemble members

In sections 3.3.1 to 3.3.4, a series of sensitivity tests for each parameter in the data assimilation system with 30 ensemble members was performed as shown in Table 1. Figure 14 shows the time series of the zonal mean temperatures at 70˚N for
500 hPa, 10 hPa, 1 hPa, and 0.1 hPa for the time period of 15 January to 20 February 2017, from respective the assimilation tests shown in Table 1 (black curves). Grey curves represent the time series from the MERRA-2. Using the time series shown in Figure 15, the RMSs of the differences and correlations between the time series of each run and MERRA-2 are calculated and are summarized in Table 3. The criteria of the major SSW were satisfied on 1 February.

The Ctrl time series is also quite similar to that of MERRA-2 in spite of the small number of ensemble members,
including drastic temperature change during the major SSW event, although a warm bias of ~4 K  is observed at 0.1 hPa before and after the cooling time period associated with the warming at 10 hPa. It is worth noting that the G5 has significant warm bias: ~4 K at 10 hPa from 31 January to 5 February during the warming event, a significant warm bias of ~10 K at 1 hPa on 23 January when a sudden temperature drop was observed, and a significant warm bias of ~10 K at 0.1 hPa before and after the cooling time period (i.e., before 27 January and after 6 February). Such significantly large biases are probably
due to the model bias, because they are not observed for the Ctrl run, in which a much larger number of observations were assimilated.





## 3.4 The effect of ensemble size and an estimate of the optimal ensemble size for the data assimilation in the middle atmosphere

The EnKF statistically estimates the forecast error covariance using ensembles. A large ensemble size (i.e., a large number of
ensemble members) is favorable because it reduces the sampling error of the covariance and improves the quality of the
analysis. However, the ensemble size has a practical limit related to the allowable computational resources. An ensemble
size of ~ 30 is usually used in the operational weather forecasting. Here, the minimum optimal ensemble size is estimated by
performing additional experiments with 90 (M90) and 200 (M200) members and comparing them with the Ctrl experiment
of 30 members (M30, Ctrl). No assimilation parameters, except for the ensemble size, are changed in the M90 and M200
experiments (see Table 1). Note that the optimal values of the assimilation parameters for the larger ensemble sizes may be
different from those for Ctrl.

Figure 15 shows the vertical profiles of the mean OmF, OmA, and $\chi^2$ for the M30 (Ctrl), M90, and M200. The
OmFs for the M90 and M200 are significantly smaller than for the M30, particularly below 0.1 hPa (by ~50%), while the
OmAs are comparable for the three runs. The difference between the OmFs of the M90 and M200 runs is small. Although
the $\chi^2$ values are ~6–17 for M30, they are ~3–4 for M90 and ~1–2 for M200, which are close to the optimal values of $\chi^2$.
The reduced values of $\chi^2$ by increasing the ensemble size are remarkable compared with those by optimizing the other
parameters (Section 3.3). However, the M90 and M200 both require such large computational costs, as already stated, that
they are not available for long-term reanalysis calculations. The time series obtained from the M90 and M200 are also shown
in Figure 14. Both time series agree well with the MERRA-2 time series and do not have even slight warm bias observed at
0.1 hPa in the Ctrl time series.

In the following, an attempt is made to estimate the minimum number of ensemble members as a function of height
using forecasts of ensemble members from the M200 experiment because 90 is a sufficient number for high quality
assimilation, judged from the similarity of the performances of the  M90 and M200 runs. Figure 16 shows the correlation
coefficient of forecasts at each longitude for a reference point of 180°E for 40°N at 10 hPa and 0.01 hPa that are computed
using 12, 25, 50, and 100 members randomly chosen from the M200 forecasts at 0600 UTC, 21 January 2017. The
longitudinal profile of the correlation coefficient for 200 members is also shown. The correlation generally reduces as the
distance from the reference point increases with large ripples for the results of ensemble sizes smaller than 50, although the
correlation profile near the reference point is expressed for all the members. Large ripples reaching ±0.6 observed for 12
members are considered spurious correlations caused by the under-sampling. In contrast, the correlation coefficients for 100
and 200 ensemble members are generally smaller than 0.2 except for a meaningfully high correlation region around the
reference point.

The RMS of the spurious correlation in the region outside the meaningful correlation region is used to estimate the
minimum optimal number of ensemble members. The RMS is examined as a function of the number of ensemble members.
Each edge longitude of the meaningful correlation region is determined where the correlation falls below 0.1 nearest the
reference point, which are 171.6°E and 171.6°W for 10 hPa and 149.1°E and 146.2°W for 0.01 hPa, for the case shown in





Figure 16. Note that the threshold value 0.1 to determine the edge longitude is arbitrary and is used as one of several possible appropriate values. The RMS of the spurious correlation is calculated by taking each longitude and latitude as a reference point. Figure 17 shows the results at 40°N and 0.01 hPa as a function of the number of ensemble members. Different curves denoted by the same thin line show the results for different longitudes. The thick curve shows mean RMS for all longitudes.

The mean RMS decreases as the number of ensemble members increases and falls to 0.1 for 91 ensemble members. Again, the choice of 0.1 is arbitrary, but from this result, we can estimate the minimum optimal ensemble size at 91. In a similar way, the minimum number of optimal ensemble members is estimated as a function of the latitude and height.

Figure 18 shows the minimum optimal number of ensemble members as a function of the height for 40°S, the equator, and 40°N. Roughly speaking, 100 members are sufficient below 80 km for all displayed latitudes except for 50 km

at the equator. The minimum optimal number of ensemble members above 80 km is larger than 100 and close to 150. From this result, more than 150 ensemble members likely give an optimal estimation of the forecast error covariance for the middle atmosphere. However, it is important to note that even if the number of ensemble members is smaller than 150, using the localization as examined in section 3.3.2 will minimize the effect of the spurious error covariance due to under-sampling, as understood from the good performance of the Ctrl assimilation using 30 ensemble members.

**4 Validation of the assimilation**

**4.1 Comparison with other reanalysis data**

This paper gives the first results of the 4D-LETKF applied to the GCM that includes the MLT region. Thus, to examine the performance of our assimilation system, the best result obtained from the M200 run among the assimilation experiments is compared with MERRA-2 as one of the standard reanalysis datasets. First, we calculated the spatial correlation of the

geopotential height anomaly from the zonal mean as a function of the pressure level and time (Figure 19). The correlation is higher than 0.9 between ~900 hPa and ~1 hPa. The top height with the high correlation varies with time. This time variation may be related to the model predictability, although such a detailed analysis is beyond the scope of the present paper. The reduction of correlation near the ground is likely due to the difference in the resolution of topography.

Next, the zonal mean zonal wind and temperature in the latitude-height section from our analysis and MERRA-2

are shown for the time periods before the major warming onset (i.e., 21–25 January) and after (i.e., 1–5 February) (Figure 20). A thick horizontal bar shows the 0.1 hPa level up to which MERRA-2 pressure level data are provided. The overall structures of the two datasets are similar: The stratopause in the northern polar region is located at a height of ~50 km (~40 km) in the early (later) period. In the northern hemisphere, an eastward jet is observed at ~63°N in the wide height range of 20–53 km in the early period. A characteristic westward jet associated with the SSW is observed in the later period in both

datasets. The spatial structure and magnitude of the westward jet are both quite similar. In the southern hemisphere, a summer westward jet is clearly observed in both sets of data. The poleward tilt with height and a maximum of ~−70 m s⁻¹ of the jet also accord well. A relatively large difference is observed around 55 km in the equatorial region. The eastward shear



with height seems much stronger in MERRA-2 than in our analysis. As the geostrophic balance does not hold in the equatorial region, it may be difficult to reproduce wind by assimilation of only temperature data. This may be the reason for the large discrepancy observed in the equatorial upper stratosphere between the two datasets.

## 4.2 Comparison with MST and meteor radar observations

The winds obtained from the M200 assimilation experiment are also compared with wind observations by meteor radars at Longyearbyen (78.2°N, 16.0°E; Hall et al., 2002) at 91 km and Kototabang (0.2°S, 100.3°E; Batubara et al., 2011) at 92 km, and by the PANSY radar at Syowa Station (69.0°S, 39.6°E) at 85 km. Note that these radar observations were not assimilated and thus can be used for validation as independent reference data. Table 4 gives a brief description of these data. Figure 21 shows the time series of zonal wind and meridional wind observed at each site (black) and corresponding 6-hourly data from our data assimilation analysis (red). A 6-h running mean was made for the time series of the radar data, although the time intervals of original data are much shorter.

Strong fluctuations with time-varying amplitudes are observed for both zonal and meridional radar winds for each station. The dominant time period is longer at Kototabang in the equatorial region than that of ~12 h at Longyearbyen in the Arctic. The amplitudes of the meridional wind fluctuations there are larger than those of zonal wind fluctuations at Kototabang. These characteristics are consistent with the wind fluctuations estimated by our assimilation system. However, there are significant differences: The time variation of the amplitudes do not accord well with observations. Differences in the phases of the oscillations between observations and estimations are sometimes small and sometimes large.

In contrast, some consistency is observed for relatively long period variations (periods longer than several days). At Longyearbyen, the slowly varying zonal wind component is slightly positive before 31 January and significantly positive from 1–6 February, while the slowly varying meridional wind tend to be significantly negative in the time period of 28–31 January. At Kototabang, the slowly varying zonal wind tends to be slightly negative before 29 January and almost zero afterward, while the slowly varying meridional wind is almost zero throughout the displayed time period. At Syowa Station, the slowly varying zonal wind tended to be negative from 23–30 January and after 2–5 February, while the slowly varying meridional wind tends to be positive from 24–31 January.

There are several plausible causes for the discrepancy in short period variations. First, there may be room to improve the model performance to reproduce such short period variations. Second, the Aura MLS provides data along a sun-synchronous orbits, and hence fluctuations associated with migrating tides may be hard for it to capture. Third, a large local increment added by the assimilation of the MLS data may cause spurious waves in the model. Fourth, there may be inertia-gravity waves with large amplitudes in the upper mesosphere (e.g., Sato et al., 2017; Shibuya et al., 2017) which cannot be captured with the relatively low-resolution GCM.





## 5 Summary and concluding remarks

A new advanced data assimilation system employing a 4D-LETKF method for the height region from the surface to the
lower thermosphere has been developed using a GCM with a very high top which we called "JAGUAR". Observation data
from NCEP PREPBUFR and Aura MLS that covered the whole neutral atmosphere up to the lower thermosphere were used
for the assimilation. The time period focused on by the present study was 10 January to 28 February 2017. This period
includes a major SSW event that occurred on 1 February in the Arctic, for which an international observation campaign for
the troposphere, stratosphere, and mesosphere was performed using a radar network.

Before optimizing parameters of the data assimilation system, the vertical profile of the horizontal dissipation and
source intensity of the non-orographic gravity wave parameterization used in JAGUAR were tuned by comparing them to
the vertical profiles of gradient winds estimated from the MLS temperature and horizontal winds observed by the PANSY
radar. The observation bias in the MLS temperature data was estimated using the SABER temperature data and subsequently
corrected.

By performing a series of sensitivity experiments, the optimal values of the other parameters were obtained for the
data assimilation system using 30 ensemble members as a practical assimilation system for the middle atmosphere. The
optimal parameter set is called the "Ctrl" experiment in Table 2. The optimized value for each parameter in the assimilation
of the atmospheric data up to the lower thermosphere was different from those used for the standard model covering the
troposphere and stratosphere. There are several possible reasons for these differences: First, the model performance is not
very mature for the MLT region. Second, the amount of observation data and observable quantities are limited for the MLT
region. Third, dominant disturbances (and dynamics as well) are different from those in the lower atmosphere. Because of
the first and second reasons, it is better to take larger gross error check coefficient in order to include a larger percentage of
the observation data. It was shown that the optimal localization length depends on the height: Smaller localization length is
better for lower heights. Thus, the best length for the middle of the model altitude range (i.e., 0.1 hPa) was employed in the
optimal parameter set. It was also shown that the inflation factor should be larger than for the standard model, although
overly large factors do not give stable ensemble spreads. A shorter assimilation window seemed better for the MLT region,
which is probably due to the dominance of short-period disturbances, such as quasi-two day waves and tides. However,
shorter assimilation windows have a problem. The amount of available observation data becomes small and the analysis thus
tends to be more reflected by the model forecasts that are not as mature as those for the lower atmosphere.

In addition, a minimum optimal number of ensemble members was examined using the results with an assimilation
system of 200 ensemble members, based on the erroneous ripple of correlation function. The minimum optimal number of
ensemble members slightly depends on the height: about 100 members below 80 km and 150 members above. It should be
noted, however, that introduction of the finite localization length to the assimilation may work to avoid spurious correlation
at distant locations even with a lower number of ensemble members than the optimal number.





The validity of the data obtained from our assimilation system was examined by comparing the MERRA-2 reanalysis dataset which has the highest top among the currently available reanalysis datasets. The correlation was greater than ~0.95 up to 1 hPa, depending on time. A comparison with radar observations in the upper mesosphere was also performed. The time variation of horizontal winds with periods longer than several days obtained from our assimilation system was consistent with the radar observations. However, the accordance of fluctuations with short wave periods,
particularly shorter than one day, was not adequate with their slight dependence on the latitude.

Nevertheless, the analysis data from our assimilation system will be useful for the study of the detailed dynamical processes in the middle atmosphere, a part of which is measured by a limited number of observation instruments. An international observation campaign by an MST radar network has been performed to capture modulation of the stratosphere and mesosphere including gravity waves initiated by the major SSW in the Arctic including the event that the present study
focused on. The low-resolution analysis product from the assimilation system developed in the present study will be used as an initial condition for a high-resolution JAGUAR model to simulate the real atmosphere including gravity waves.

In future work, we plan to use more observation data in the middle atmosphere for the assimilation. These include satellite data, such as temperature observation data from the SABER, radiance data from the SSMIS, and Global Navigation Satellite System (GNSS) radio occultation data, and also wind data from radars in the mesosphere. We also plan to examine
the impact of assimilating these data with observation system simulation experiments using simulation data from a high-resolution GCM.

**Code availability**

The source codes for the data assimilation are available for the editor and reviewers. The copyright of the code for LETKF belongs to Takemasa Miyoshi, and the related code can be accessed from https://github.com/takemasa-miyoshi/letkf (last
access: 9 October 2019).

**Data availability**

Meteor radar data from Kototabang are available at the Inter-university Upper atmosphere Global Observation NETwork (IUGONET) site (http://www.iugonet.org). Meteor radar data from Longyearbyen are available by request from the National Institute of Polar Research by contacting Masaki Tsutsumi (tutumi@nipr.ac.jp). The PANSY radar observational data are
available at the project website, http://pansy.eps.s.u-tokyo.ac.jp. NCEP PREPBUFR data are available from https://rda.ucar.edu/datasetsds337.0/ (accessed 26 Apr 2017). Aura MLS data, which are compiled and archived by NASA, were also used for the data assimilation (available from https://acdisc.gesdisc.eosdis.nasa.gov/data/Aura_MLS_Level2/, accessed 21 Jul 2019).





## Appendix A

Here we show the equivalence of the two methods, namely, the calculation of time development after the data assimilation and the 4D data assimilation with the calculation of time development. The case of $j2 = j1 + 1$, that is, $\mathbf{x}_{j2} = \mathbf{M}\mathbf{x}_{j1}$ is considered. In the first method, the analysis $\mathbf{x}_{j1}^a$ and the analysis error covariance matrix $\mathbf{P}_{j1}^a$ at the time step $j1$ are written using (7) and (8):

$$\mathbf{x}_{j1}^a = \mathbf{x}_{j1}^f + \mathbf{P}_{j1}^a \mathbf{H}^{\mathrm{T}} \mathbf{R}^{-1}(\mathbf{y} - \mathbf{H}\mathbf{x}_{j1}^f), \tag{A1}$$

$$\mathbf{P}_{j1}^a = \left(\mathbf{I} - \mathbf{P}_{j1}^f \mathbf{H}^{\mathrm{T}}\left(\mathbf{H}\mathbf{P}_{j1}^f \mathbf{H}^{\mathrm{T}} + \mathbf{R}\right)^{-1}\mathbf{H}\right)\mathbf{P}_{j1}^f. \tag{A2}$$

Using the model forecast matrix $\mathbf{M}$, $\mathbf{x}_{j2}^a$ and $\mathbf{P}_{j2}^a$ at the next time step $j2$ are obtained:

$$\mathbf{x}_{j2}^a = \mathbf{M}\mathbf{x}_{j1}^a = \mathbf{M}\mathbf{x}_{j1}^f + \mathbf{M}\mathbf{P}_{j1}^a \mathbf{H}^{\mathrm{T}} \mathbf{R}^{-1}(\mathbf{y} - \mathbf{H}\mathbf{x}_{j1}^f), \tag{A3}$$

$$\mathbf{P}_{j2}^a = \mathbf{M}\mathbf{P}_{j1}^a \mathbf{M}^{\mathrm{T}} = \mathbf{M}\left(\mathbf{I} - \mathbf{P}_{j1}^f \mathbf{H}^{\mathrm{T}}\left(\mathbf{H}\mathbf{P}_{j1}^f \mathbf{H}^{\mathrm{T}} + \mathbf{R}\right)^{-1}\mathbf{H}\right)\mathbf{P}_{j1}^f \mathbf{M}^{\mathrm{T}}. \tag{A4}$$

In the second method, the analysis $\mathbf{x}_{j2}^a$ is written as

$$\mathbf{x}_{j2}^a = \mathbf{x}_{j2}^f + \mathbf{P}_{j2}^a \mathbf{H}'^{\mathrm{T}} \mathbf{R}^{-1}\left(\mathbf{y} - \mathbf{H}'^{\mathbf{x}_{j2}^f}\right), \tag{A5}$$

where $\mathbf{H}'$ is the observation operator at the time step $j2$ and is related to $\mathbf{H}$:

$$\mathbf{H}' = \mathbf{H}\mathbf{M}^{-1}. \tag{A6}$$

By substituting this formula into Eq. (A5), the analysis $\mathbf{x}_{j2}^a$ is written as

$$\begin{aligned}
\mathbf{x}_{j2}^a &= \mathbf{M}\mathbf{x}_{j1}^f + \mathbf{P}_{j2}^a (\mathbf{H}\mathbf{M}^{-1})^{\mathrm{T}} \mathbf{R}^{-1}(\mathbf{y} - \mathbf{H}\mathbf{M}^{-1}\mathbf{x}_{j2}^f) \\
&= \mathbf{M}\mathbf{x}_{j1}^f + \mathbf{P}_{j2}^f (\mathbf{M}^{-1})^T \mathbf{H}^{\mathrm{T}} \mathbf{R}^{-1}(\mathbf{y} - \mathbf{H}\mathbf{M}^{-1}\mathbf{x}_{j2}^f) \\
&= \mathbf{M}\mathbf{x}_{j1}^f + \mathbf{M}\mathbf{P}_{j1}^f \mathbf{H}^{\mathrm{T}} \mathbf{R}^{-1}(\mathbf{y} - \mathbf{H}\mathbf{x}_{j1}^f) \\
&= \mathbf{M}\mathbf{x}_{j1}^a,
\end{aligned}$$

which is identical to Eq. (A3).

Similarly, the analysis error covariance $\mathbf{P}_{j2}^a$ is written as

$$\mathbf{P}_{j2}^a = \left(\mathbf{I} - \mathbf{P}_{j2}^f \mathbf{H}'^{\mathrm{T}}\left(\mathbf{H}'\mathbf{P}_{j2}^f \mathbf{H}'^{\mathrm{T}} + \mathbf{R}\right)^{-1}\mathbf{H}'\right)\mathbf{P}_{j2}^f \quad \text{(A7)},$$

which is transformed using Eq. (A6) to

$$\begin{aligned}
\mathbf{P}_{j2}^a &= \left(\mathbf{I} - \mathbf{P}_{j2}^f (\mathbf{H}\mathbf{M}^{-1})^{\mathrm{T}}\left(\mathbf{H}\mathbf{M}^{-1}\mathbf{P}_{j2}^f (\mathbf{H}\mathbf{M}^{-1})^{\mathrm{T}} + \mathbf{R}\right)^{-1}\mathbf{H}\mathbf{M}^{-1}\right)\mathbf{M}\mathbf{P}_{j1}^f \mathbf{M}^{\mathrm{T}} \\
&= \left(\mathbf{I} - \mathbf{P}_{j2}^f \mathbf{M}^{-\mathrm{T}}\mathbf{H}^{\mathrm{T}}\left(\mathbf{H}\mathbf{M}^{-1}\mathbf{P}_{j2}^f \mathbf{M}^{-\mathrm{T}}\mathbf{H}^{\mathrm{T}} + \mathbf{R}\right)^{-1}\mathbf{H}\mathbf{M}^{-1}\right)\mathbf{M}\mathbf{P}_{j1}^f \mathbf{M}^{\mathrm{T}} \\
&= \left(\mathbf{I} - \mathbf{M}\mathbf{P}_{j1}^f \mathbf{H}^{\mathrm{T}}\left(\mathbf{H}\mathbf{P}_{j1}^f \mathbf{H}^{\mathrm{T}} + \mathbf{R}\right)^{-1}\mathbf{H}\mathbf{M}^{-1}\right)\mathbf{M}\mathbf{P}_{j1}^f \mathbf{M}^{\mathrm{T}} \\
&= \mathbf{M}\left(\mathbf{I} - \mathbf{P}_{j1}^f \mathbf{H}^{\mathrm{T}}\left(\mathbf{H}\mathbf{P}_{j1}^f \mathbf{H}^{\mathrm{T}} + \mathbf{R}\right)^{-1}\mathbf{H}\right)\mathbf{P}_{j1}^f \mathbf{M}^{\mathrm{T}} \\
&= \mathbf{M}\mathbf{P}_{j1}^a \mathbf{M}^{\mathrm{T}},
\end{aligned}$$






which is identical to Eq. (A4).

These relations can be derived for any $j2$ other than $j2 = j1 + 1$.

**Author contribution**

D. Koshin and K. Sato designed the experiments and D. Koshin carried them out. S. Watanabe developed the forecast model (JAGUAR) code. K. Miyazaki implemented the data assimilation module into JAGUAR. K. Sato and D. Koshin prepared the manuscript with contributions from all the co-authors.

**Acknowledgements**

We greatly appreciate Takemasa Miyoshi, Tomoyuki Higuchi, and Hiromichi Nagao for their fruitful discussion and also
Masaki Tsutsumi and Chris Hall for providing the meteor radar data from Longyearbyen. The data assimilation experiments were performed using the Japan Agency for Marine-Earth Science and Technology (JAMSTEC)'s Data Analyzer (DA) system. This work is supported by JST CREST Number JPMJCR1663, Japan. Part of this work was performed at the Jet Propulsion Laboratory, California Institute of Technology, under a contract with NASA. PANSY is a multi-institutional project with core members from the University of Tokyo, National Institute of Polar Research, and Kyoto University. The
PANSY radar measurements at Syowa Station are operated by the Japanese Antarctic Research Expedition (JARE).

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





**Table 1: Setting parameters. Bold faces show the difference from the control (the first line).**

|  | Diffusion coeff. | GWP source intensity | Gross error check for MLS | Localization length (km) | Inflation coeff. (%) | Window length (h) | Number of members |
|---|---|---|---|---|---|---|---|
| Ctrl | **B** | 0.7 | 20 | 600 | 15 | 6 | 30 |
| DC | **C** | 0.7 | 20 | 600 | 15 | 6 | 30 |
| P0.5 | B | **0.5** | 20 | 600 | 15 | 6 | 30 |
| P1 | B | **1.0** | 20 | 600 | 15 | 6 | 30 |
| G5 | B | 0.7 | **5** | 600 | 15 | 6 | 30 |
| L300 | B | 0.7 | 20 | **300** | 15 | 6 | 30 |
| L1000 | B | 0.7 | 20 | **1000** | 15 | 6 | 30 |
| I7 | B | 0.7 | 20 | 600 | **7** | 6 | 30 |
| I25 | B | 0.7 | 20 | 600 | **25** | 6 | 30 |
| W3 | B | 0.7 | 20 | 600 | 15 | **3** | 30 |
| W12 | B | 0.7 | 20 | 600 | 15 | **12** | 30 |
| M90 | B | 0.7 | 20 | 600 | 15 | 6 | **90** |
| M200 | B | 0.7 | 20 | 600 | 15 | 6 | **200** |





**Table 2: The localization length dependence of the root mean square (RMS) difference [K] between the analysis temperature and the MLS temperature observation. The averaged data for the time period of 12 January to 20 February 2017 is shown.**

| Height (hPa) | L300 | L600 (Ctrl) | L1000 |
|---|---|---|---|
| 0.005 | 10.1 | 8.6 | 8.3 |
| 0.01 | 11.6 | 9.2 | 9.0 |
| 0.1 | 6.8 | 5.5 | 6.4 |
| 1 | 3.8 | 4.1 | 5.9 |
| 10 | 2.1 | 2.6 | 3.6 |




**Table 3: The bias of the time series of zonal mean temperature at 70 °N for the time period from 15 January to 20 February 2017 from each assimilation experiment and from MERRA-2 (see Figure 14). The RMS of the differences between the time series of each experiment and MERRA-2, as well as the correlation (Corr) between the time series from each experiment and from MERRA-2, are also shown. Results for (a) 500 hPa, (b) 10 hPa, (c) 1 hPa, and (d) 0.1 hPa are shown.**

|       | (a) 500hPa | | (b) 10hPa | | (c) 1.0hPa | | (d) 0.1hPa | |
|-------|------|------|------|------|------|------|------|------|
|       | RMS | Corr | RMS | Corr | RMS | Corr | RMS | Corr |
| Ctrl  | 0.808 | 0.928 | 1.635 | 0.994 | 3.358 | 0.954 | 4.069 | 0.959 |
| DC    | 0.787 | 0.914 | 1.602 | 0.995 | 3.888 | 0.944 | 4.566 | 0.944 |
| P0.5  | 1.009 | 0.912 | 1.729 | 0.989 | 3.630 | 0.951 | 4.599 | 0.943 |
| P1    | 1.062 | 0.926 | 1.813 | 0.993 | 4.579 | 0.926 | 5.051 | 0.957 |
| G5    | 0.826 | 0.929 | 2.295 | 0.990 | 5.140 | 0.923 | 8.870 | 0.848 |
| L300  | 0.987 | 0.951 | 1.535 | 0.995 | 3.030 | 0.956 | 8.014 | 0.949 |
| L1000 | 1.700 | 0.397 | 2.385 | 0.980 | 5.941 | 0.915 | 3.812 | 0.951 |
| I7    | 1.023 | 0.810 | 1.694 | 0.995 | 3.548 | 0.965 | 7.028 | 0.970 |
| I25   | 0.835 | 0.956 | 2.022 | 0.986 | 4.593 | 0.922 | 4.239 | 0.900 |
| W3    | 1.005 | 0.915 | 1.809 | 0.988 | 3.506 | 0.950 | 3.385 | 0.946 |
| W12   | 1.321 | 0.720 | 2.110 | 0.992 | 3.759 | 0.946 | 6.853 | 0.956 |
| M90   | 0.794 | 0.976 | 1.570 | 0.996 | 2.381 | 0.970 | 2.408 | 0.973 |
| M200  | 0.879 | 0.961 | 1.500 | 0.997 | 1.932 | 0.977 | 2.333 | 0.975 |






**Table 4: Location, radar type, and vertical resolution used for the comparison with the analysis. "MST radar" stands for the Mesosphere-Stratosphere-Troposphere radar.**

| Station | Radar type | Vertical resolution | Time interval | Organization |
|---|---|---|---|---|
| Longyearbyen (78.2˚N, 16.0˚E) | Meteor radar | 2km | 30 min | NIPR |
| Kototabang (0.2˚S, 100.3˚E) | Meteor radar | 2km | 1 h | Kyoto University |
| Syowa Station (69.0˚S, 39.6˚E) | MST radar (The PANSY radar) | 300m | 30 min | The University of Tokyo/ NIPR |



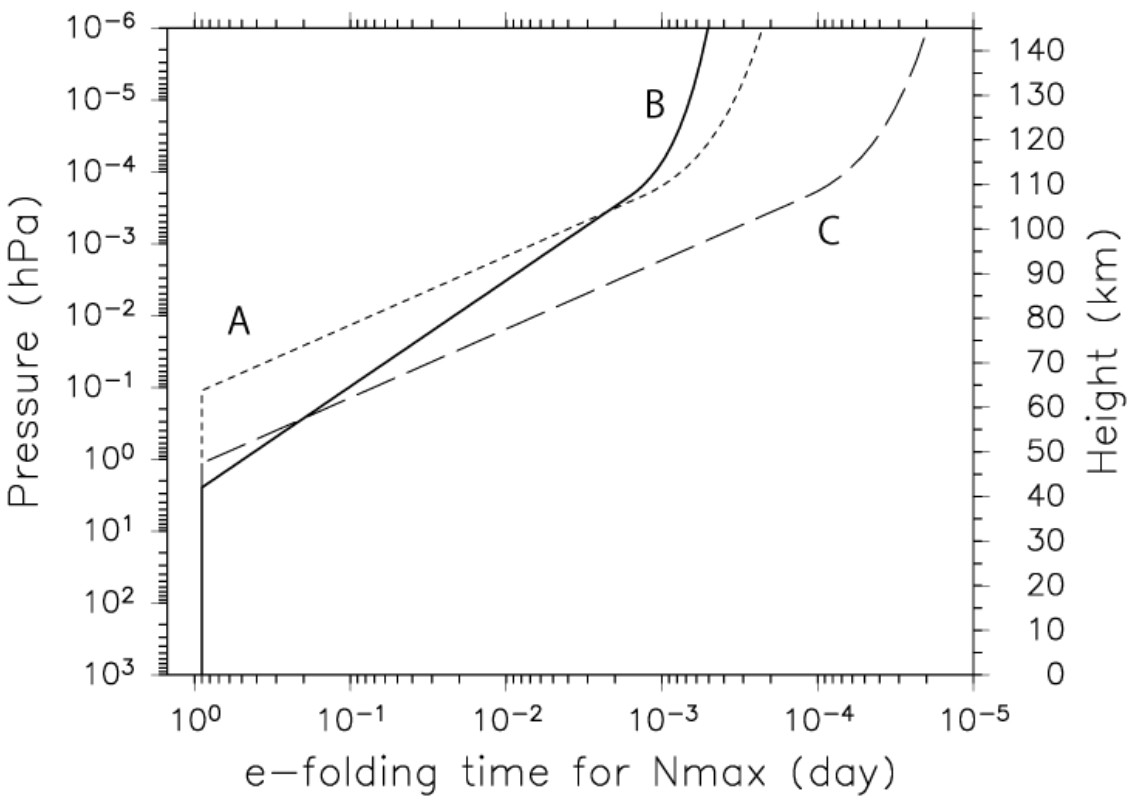

**Figure 1: The vertical profiles of the horizontal diffusion coefficients given in the forecast model. The profile B was used for the data assimilation.**





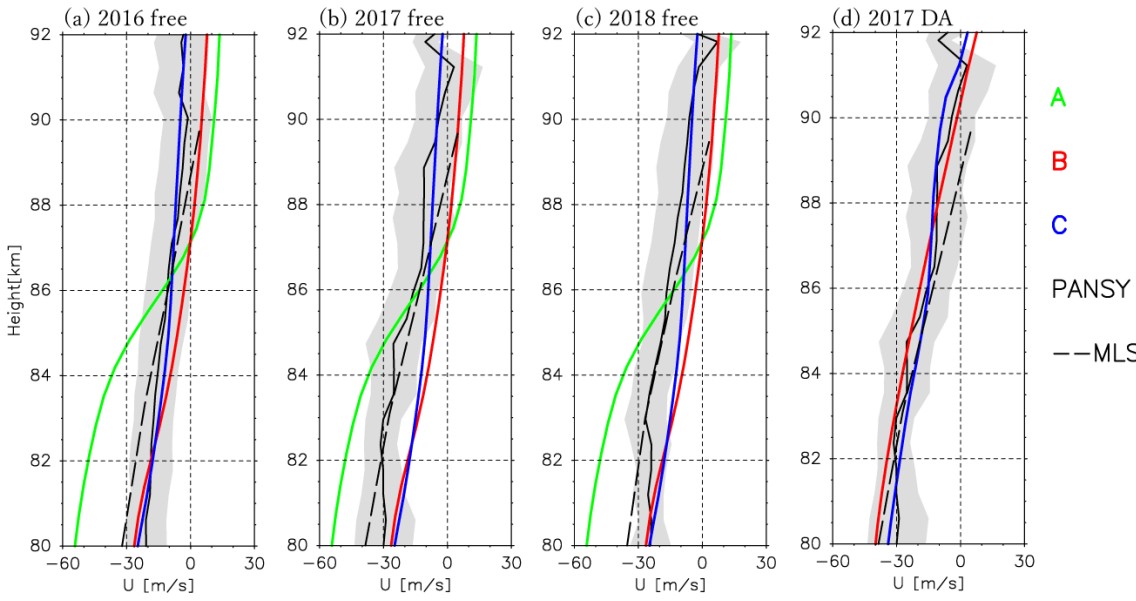

**Figure 2: Vertical profiles of the zonal mean zonal wind averaged for the time period of 12 January to 20 February from free runs with different horizontal diffusions (A: green curves, B: red curves, and C: blue curves) for (a) 2016, (b) 2017, and (c) 2018. PANSY radar and MLS observations are also shown by black solid curves and dashed curves, respectively. Gray shading denotes the range of standard deviation for the PANSY radar observation during the time period. (d) Results of the data assimilation with the Ctrl parameter set for 2017.**





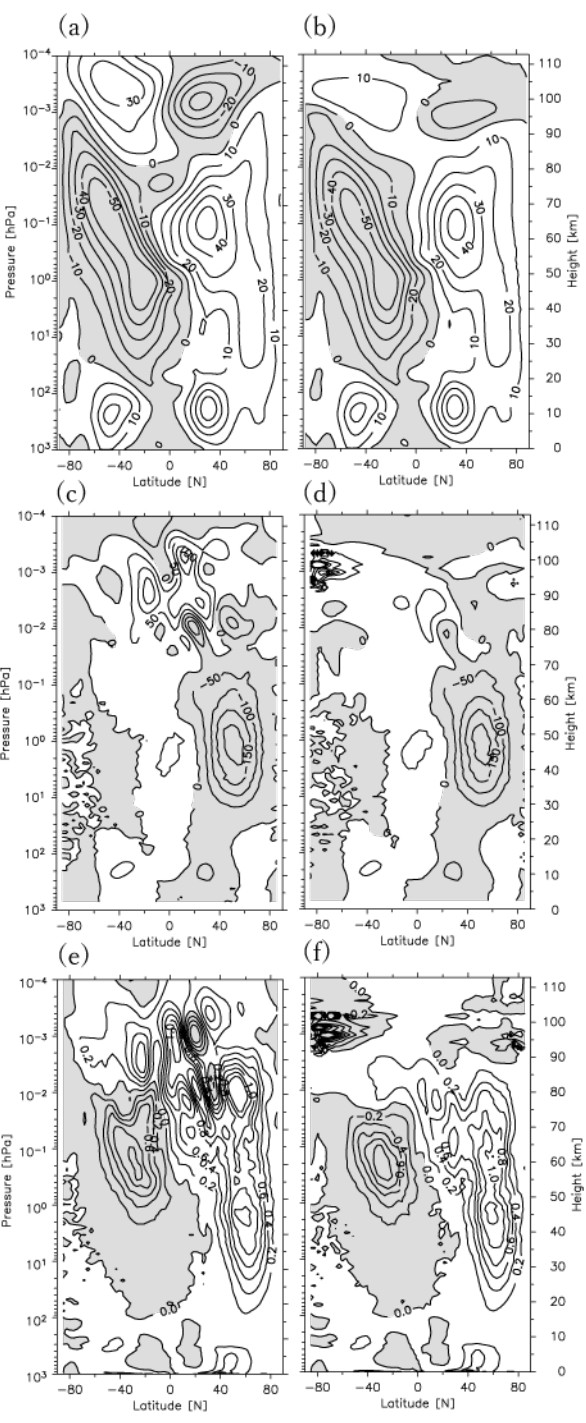

**Figure 3: The meridional cross sections of the zonal mean zonal wind [(a) and (b)], the meridional component of E-P flux [(c) and (d)], and the vertical component of E-P flux [(e) and (f)]. (a), (c), and (e) [(b), (d), and (f)] obtained using the results of the data assimilation for DB (Ctrl) [DC], and averaged for the time period of 12 January to 20 February 2017. Contour intervals are 10 m s$^{-1}$ for (a) and (b), 50 m$^2$s$^{-2}$ for (c) and (d), and 0.2 m$^2$s$^{-2}$ for (e) and (f).**



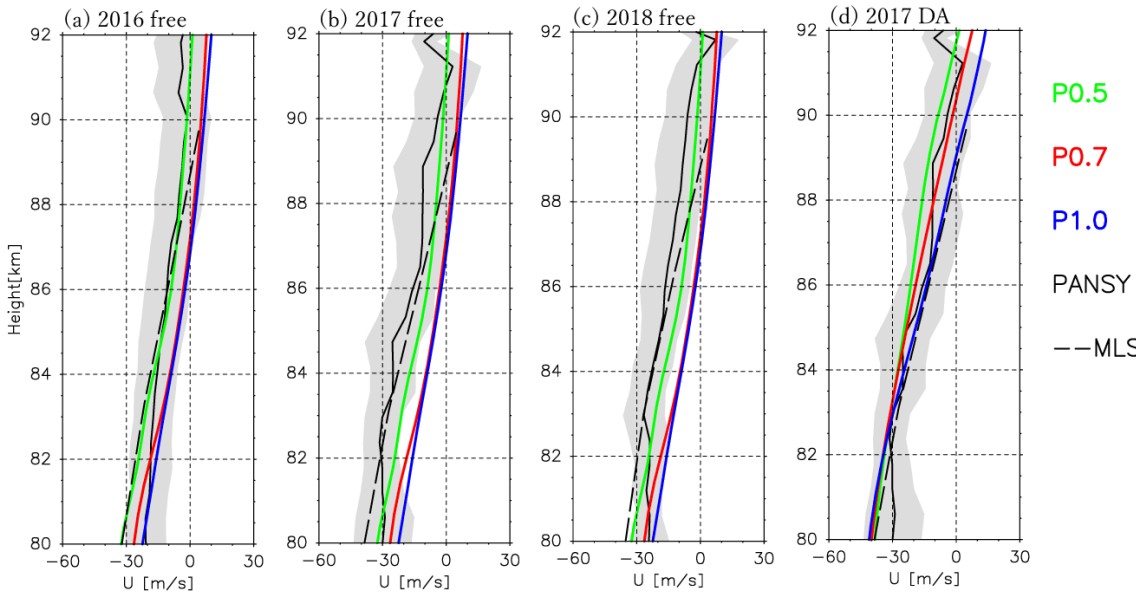

**Figure 4: The vertical profiles of the zonal mean zonal wind averaged for the time period of 12 January to 20 February from free runs with gravity-wave parameterization of different source intensities (P0.5: green curves, P0.7: red curves, and P1.0: blue curves) for (a) 2016, (b) 2017, and (c) 2018. PANSY radar and MLS observations are also shown by black solid curves and dashed curves respectively. Gray shading denotes the range of standard deviation for the PANSY radar observation during the time period. (d) Results of the data assimilation with the Ctrl parameter set for 2017.**

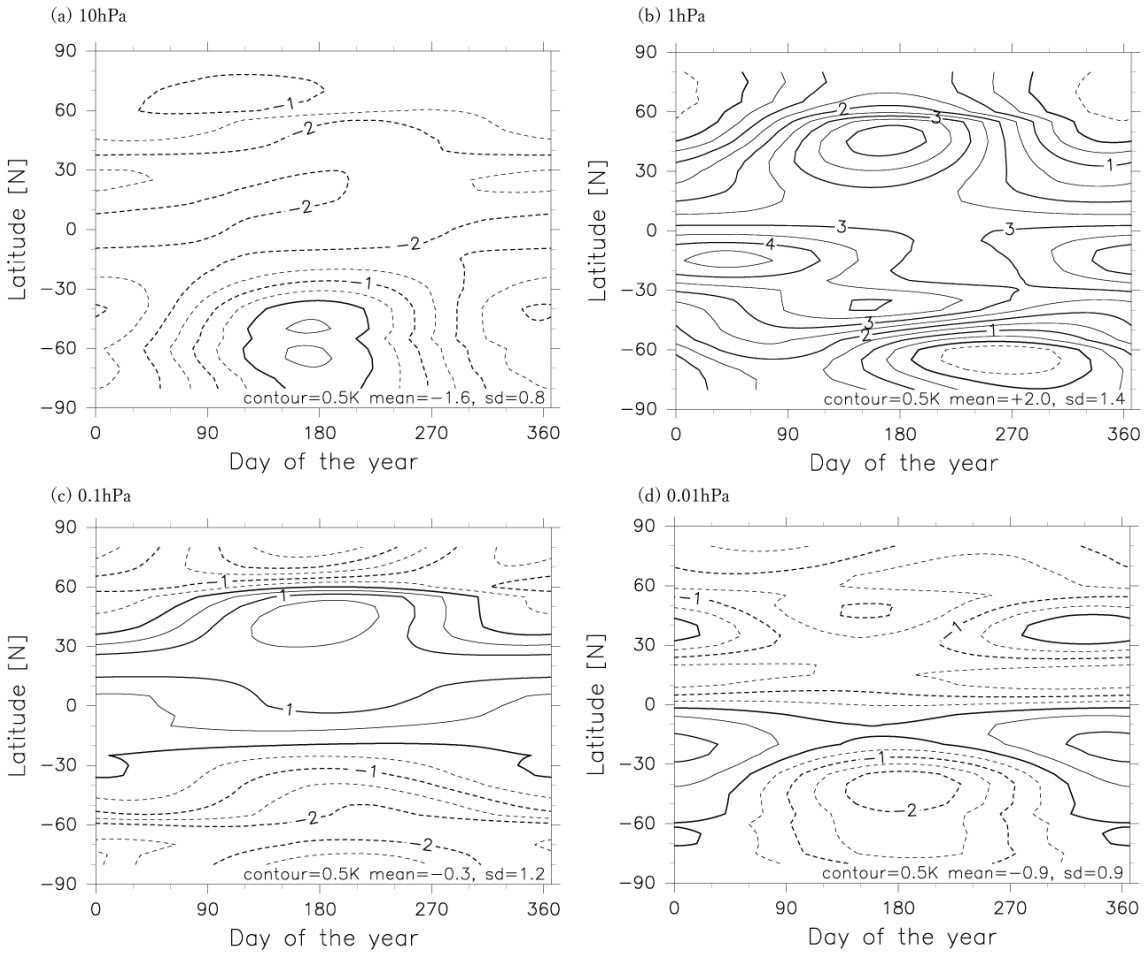

**Figure 5: The day-latitude section of MLS bias at (a) 10 hPa, (b) 1 hPa, (c) 0.1 hPa, and (d) 0.01 hPa. The contour intervals are 0.5 K.**





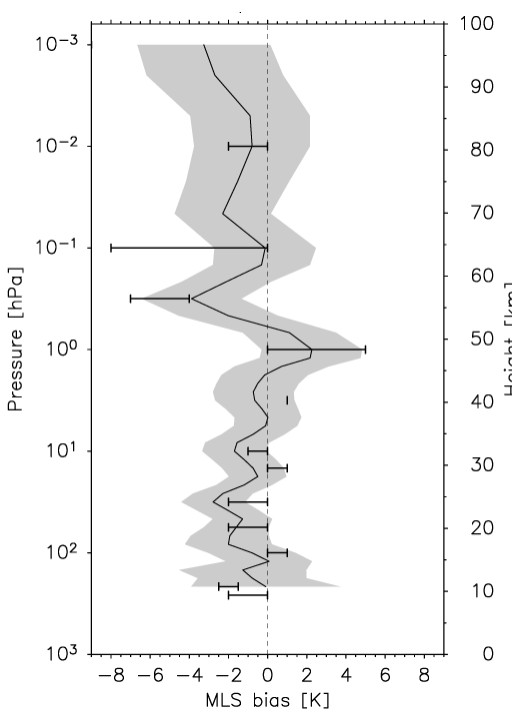

**Figure 6: The vertical profile of the global average of the Aura MLS temperature bias (the solid black curve) with standard deviation (gray shading). Error bars denote reported bias (Livesey et al., 2018).**



**Figure 7: The meridional cross sections of zonal mean temperature [(a) and (b)] and zonal wind [(e) and (f)]. The results (a) and (e) [(b) and (f)] from the assimilating Aura MLS data without [with] bias correction which are averaged for the time period of 12 January to 20 February 2017. (c) [(g)] The difference between (a) and (b) [(e)–(f)]. (d) The corrected bias for the same time period. Contour intervals are 10 K for (a) and (b), 2 K for (c) and (d), 10 m s⁻¹ for (e) and (f), and 5 m s⁻¹ for (g).**





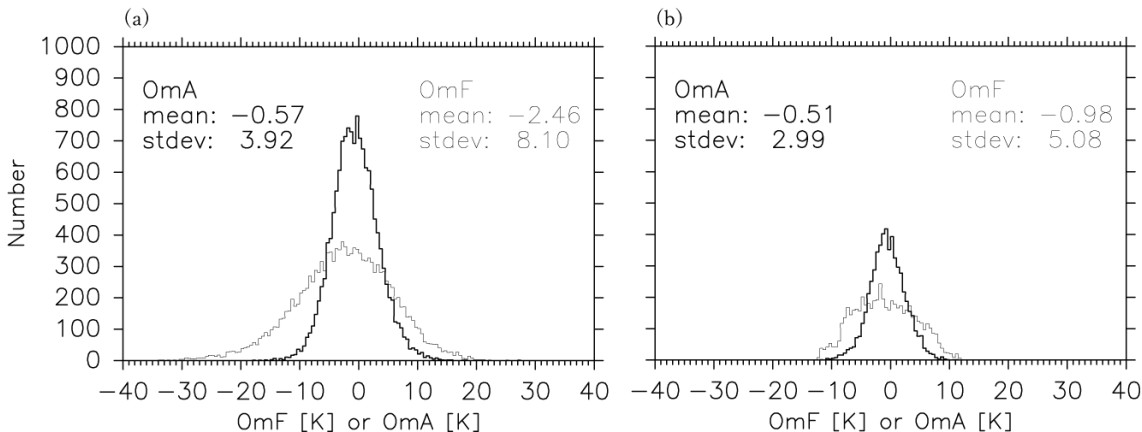

**Figure 8: Histogram of the OmF (thin curves) and OmA (thick curves) at 0.1 hPa from (a) the G20 and (b) the G5 for the time period of 12 January to 20 February 2017.**






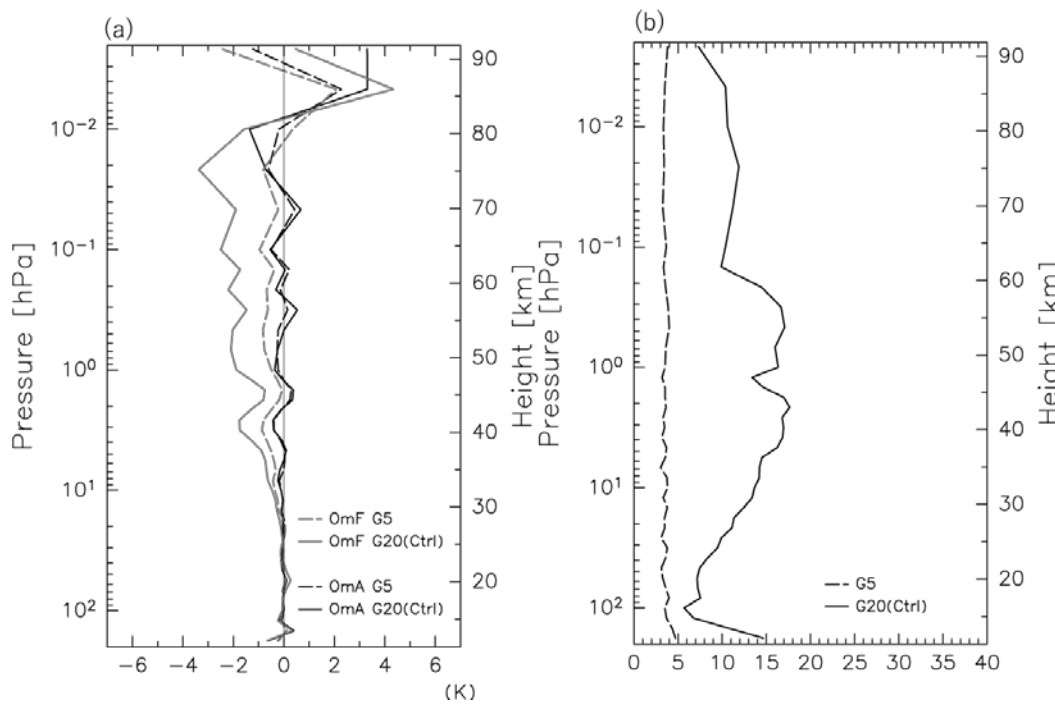

**Figure 9: The vertical profiles of the global mean (a) OmF and OmA, and (b) $\chi^2$. The gray (black) curves denote the OmF (OmA) in (a). Dashed and solid curves denote results from the G5 and G20 (Ctrl), respectively. Plotted are an average for the time period of 12 January to 20 February 2017.**





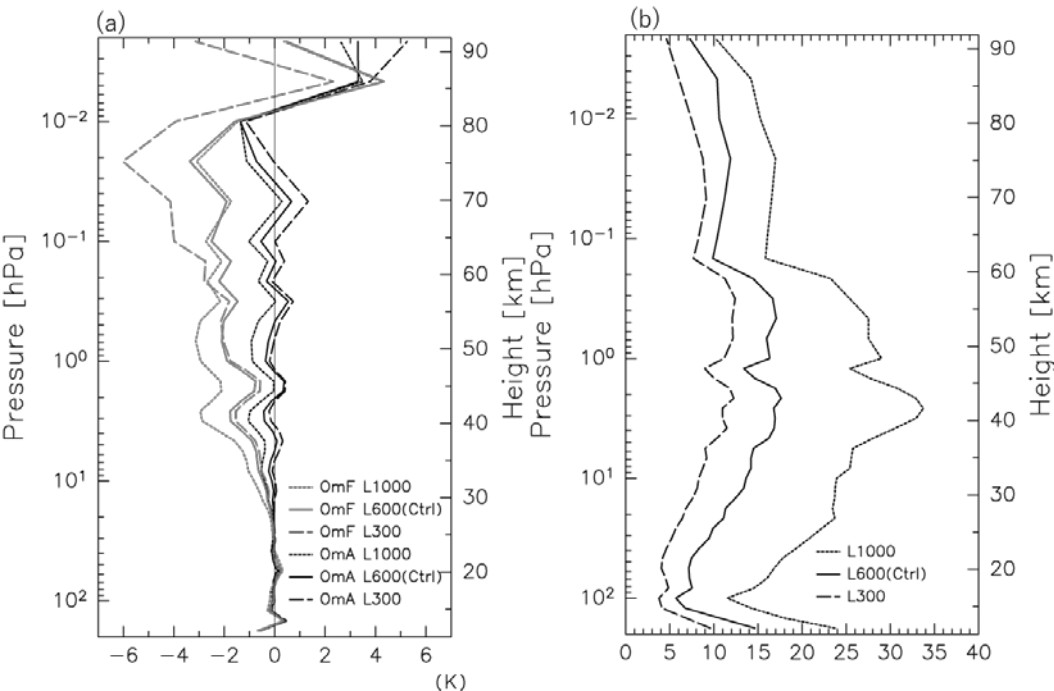

**Figure 10: The vertical profiles of the global mean (a) OmF and OmA, and (b) $\chi^2$. The bray (black) curves denote the OmF (OmA) in (a). Dashed, solid, and dotted curves denote results from L300, L600 (Ctrl), and L1000, respectively. Plotted are an average for the time period of 12 January to 20 February 2017.**


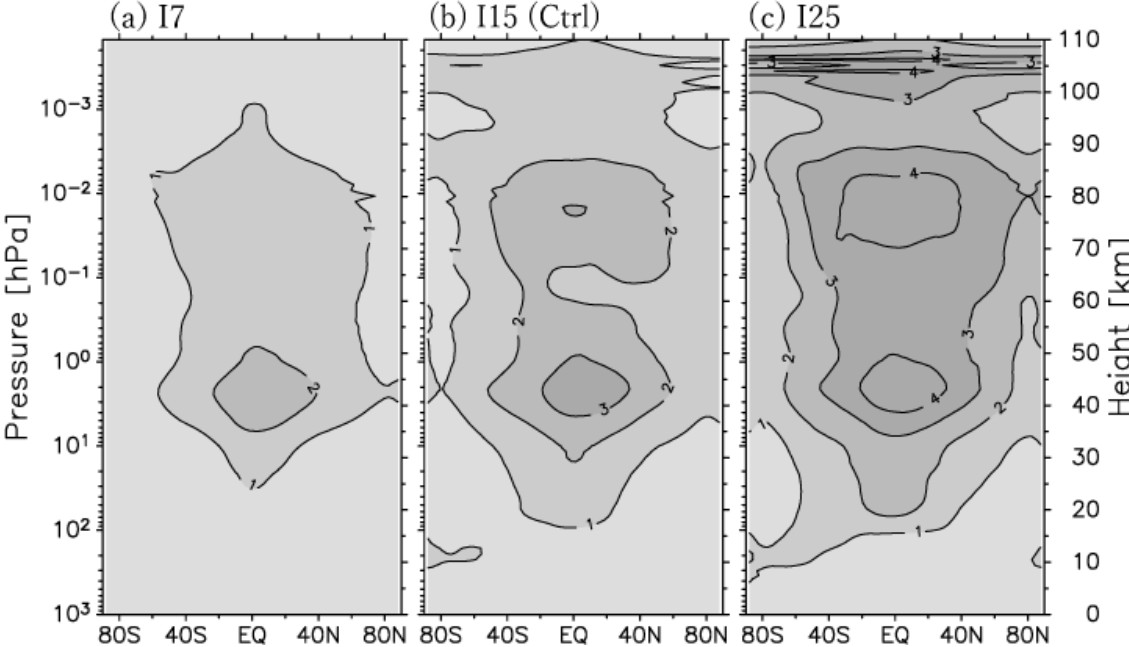

**Figure 11: The meridional cross section of the zonal mean ensemble spread of temperature [K] for (a) I7, (b) I15 (Ctrl), and (c) I25 averaged for the time period of 12 January to 20 February 2017.**






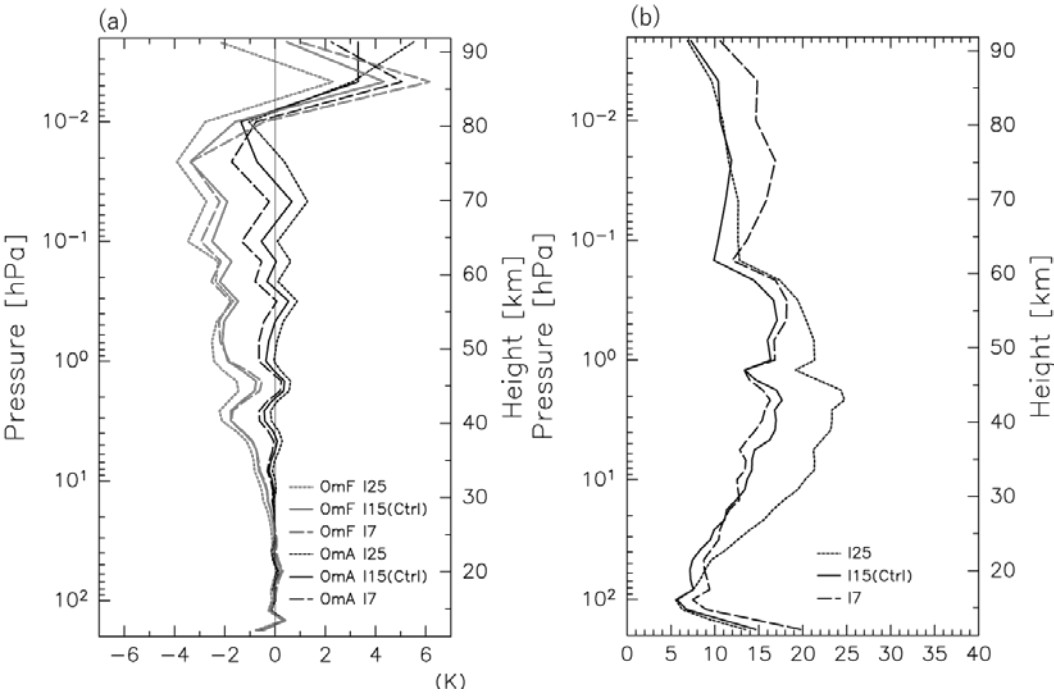

**Figure 12: The vertical profiles of the global mean (a) OmF and OmA, and (b) $\chi^2$. The gray (black) curves denote the OmF (OmA) in (a). Dashed, solid, and dotted curves denote results from I7, I15 (Ctrl), and I25, respectively. Plotted are an average for the time period of 12 January to 20 February 2017.**





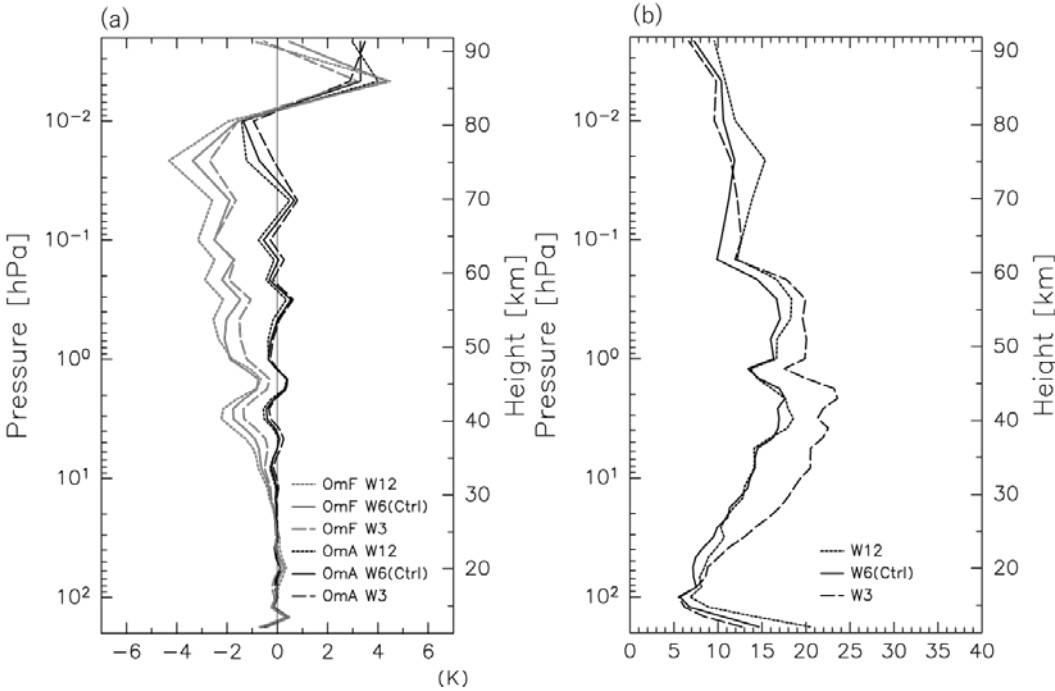

Figure 13: The vertical profiles of the global mean (a) OmF and OmA, and (b) $\chi^2$. The gray (black) curves denote the OmF (OmA) in (a). Dashed, solid, and dotted curves denote results from W3, W6 (Ctrl), and W12, respectively. Plotted are an average for the time period of 12 January to 20 February 2017.








**Figure 14: The time series of the zonal mean temperature at 70˚N for (a) 500hPa, (b) 10hPa, (c) 1hPa, and (d) 0.1hPa. The black curves show the results from each parameter setting (see Table 1). The right axis is given for the result of Ctrl, and the other curves are vertically shifted by (a) 5K, (b) 15K, (c) 10K, and (d) 15K. The gray curves shows the time series calculated using MERRA-2 as a reference.**

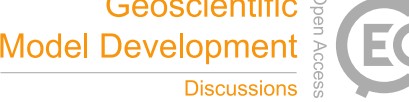

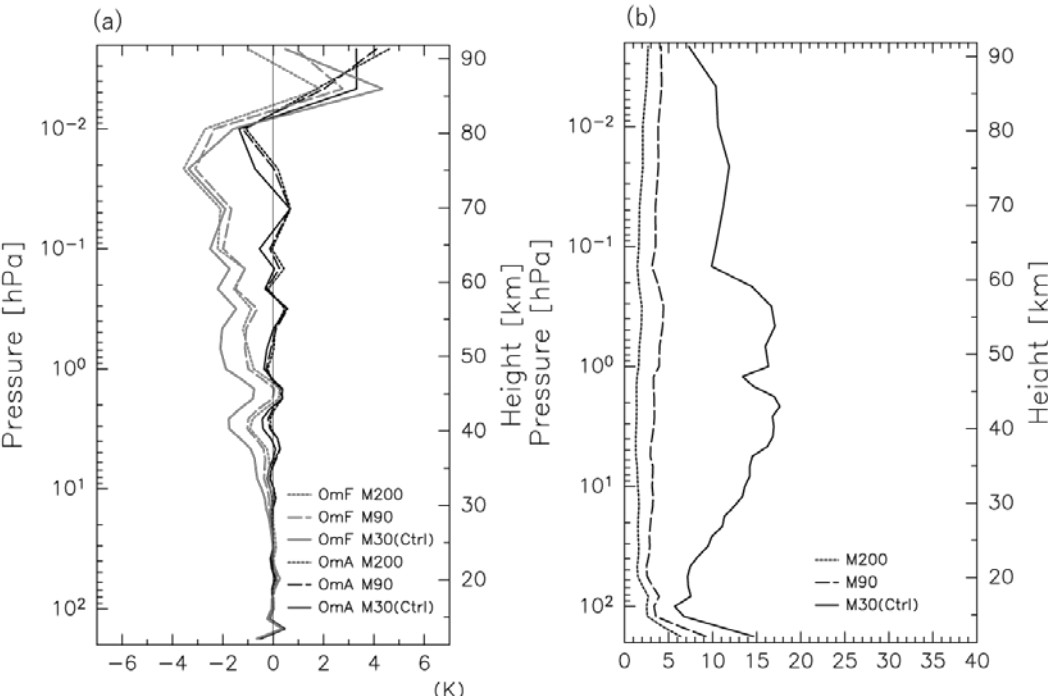


**Figure 15: The vertical profiles of the global mean (a) OmF and OmA, and (b) $\chi^2$. The gray (black) curves denote the OmF (OmA) in (a). Dashed, solid, and dotted curves denote results from M30 (Ctrl), M90, and M200, respectively. Plotted are an average for the time period of 12 January to 20 February 2017.**



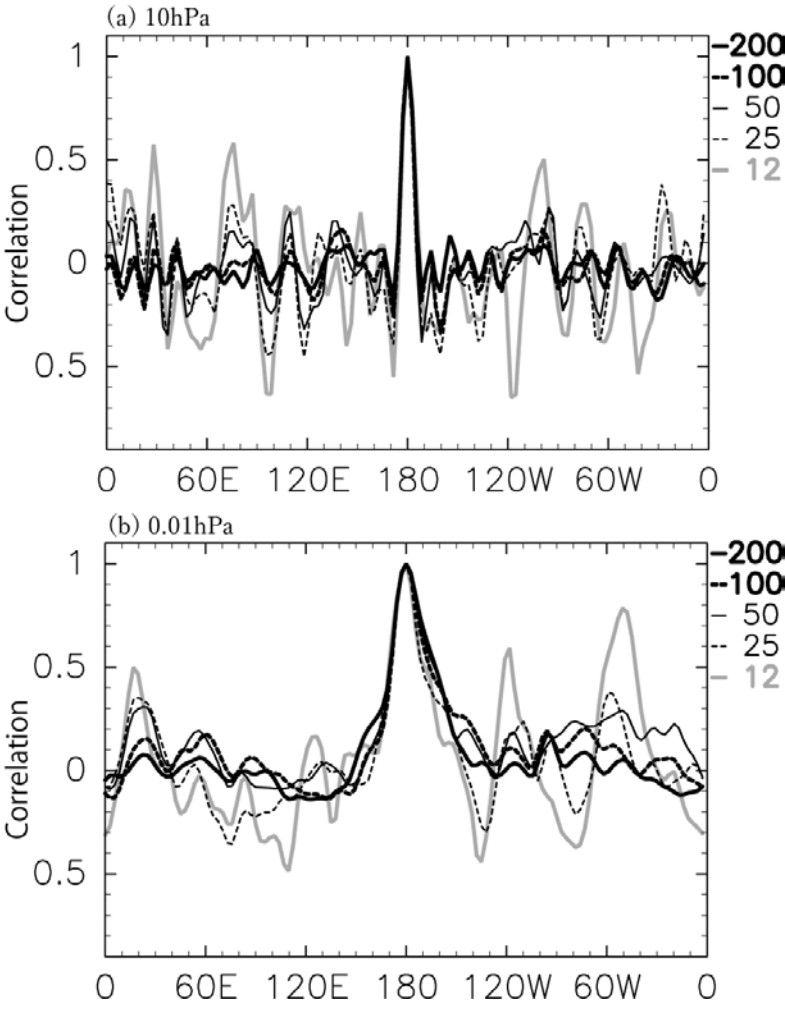

**Figure 16: The ensemble correlation of temperature at 40°N, 10hPa (a) and 0.01hPa (b). Each curve shows the results of the 200 (a thick solid curve), 100 (thick dashed), 50 (thin solid), 25 (thin dashed), and 12 (thick grey) ensembles.**





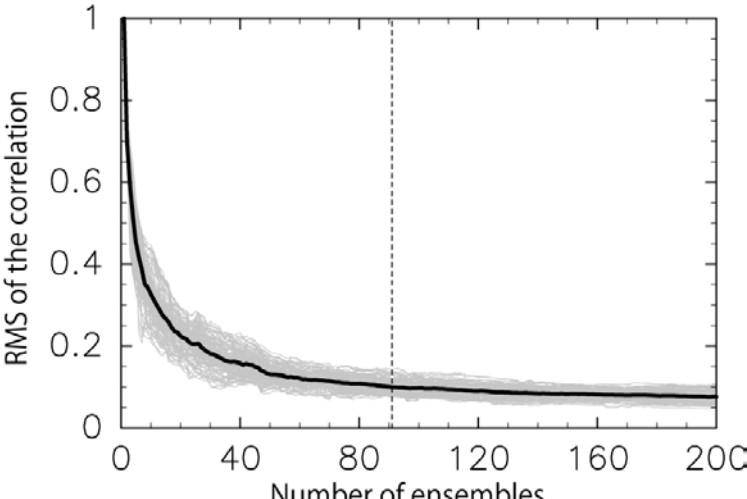

**Figure 17: The RMS of spurious correlation at 40˚N, 0.01 hPa, 0600 UTC, 21 January 2017, as a function of the number of ensembles. The gray curves show the results of respective longitude, and a black curve shows the average. A dashed line shows the number of members for which the mean RMS is 0.1 (i.e., 91).**




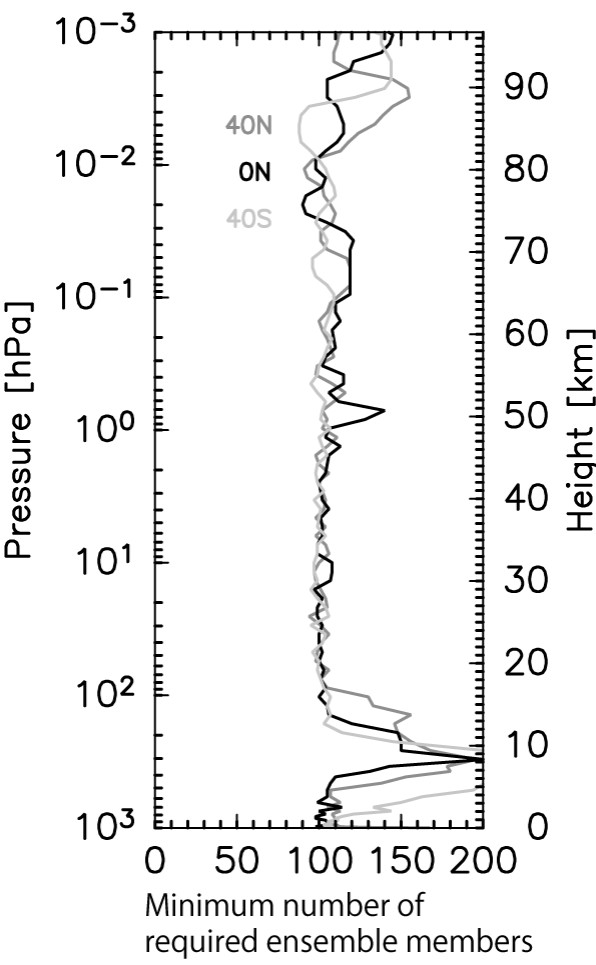

**Figure 18: The vertical profiles of the minimum number of required ensemble members that were estimated from the RMS of spurious correlation. The black, dark gray, and light gray curves show the results for the equator, 40˚N, and 40˚S, respectively.**



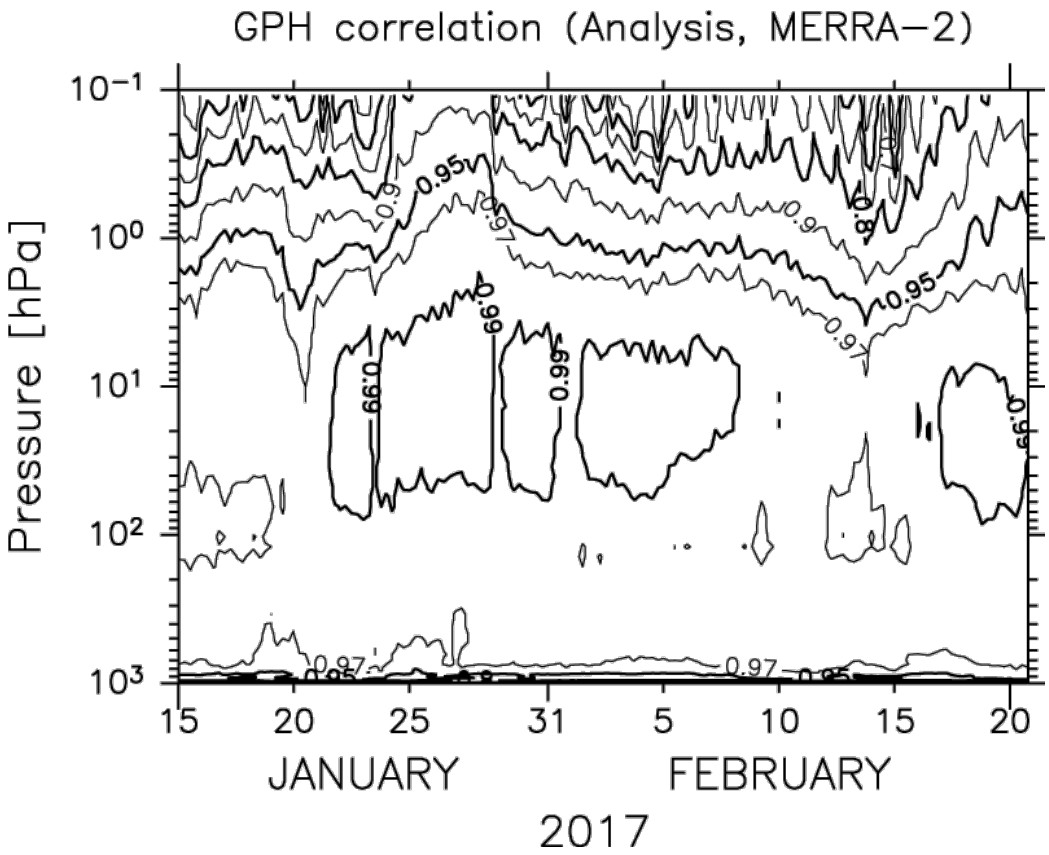

**Figure 19: The zonal mean of the spatial correlation of the geopotential height anomaly from the zonal mean between the analysis (M200) and MERRA-2. Contours of 0.5, 0.6, 0.7, 0.8, 0.9, 0.95, 0.97, and 0.99 are shown.**








**Figure 20: The meridional cross section of the zonal mean temperature [(a), (b), (e), (f), (i), and (j)] and zonal wind [(c), (d), (g), and (h)]. (a), (c), (e), (g), and (i) [(b), (d), (f), (h), and (j)] averaged for the time period of 21–25 January 2017 [1–5 February 2017]. (a), (b), (c), and (d) [(e), (f), (g), and (h)] are the results of the data assimilation (M200), and (i) and (j) are the results of the Aura MLS data. The contour intervals are 10 K for (a), (b), (e), (f), (i), and (j) and 10 m s$^{-1}$ for (c), (d), (g), and (h).**



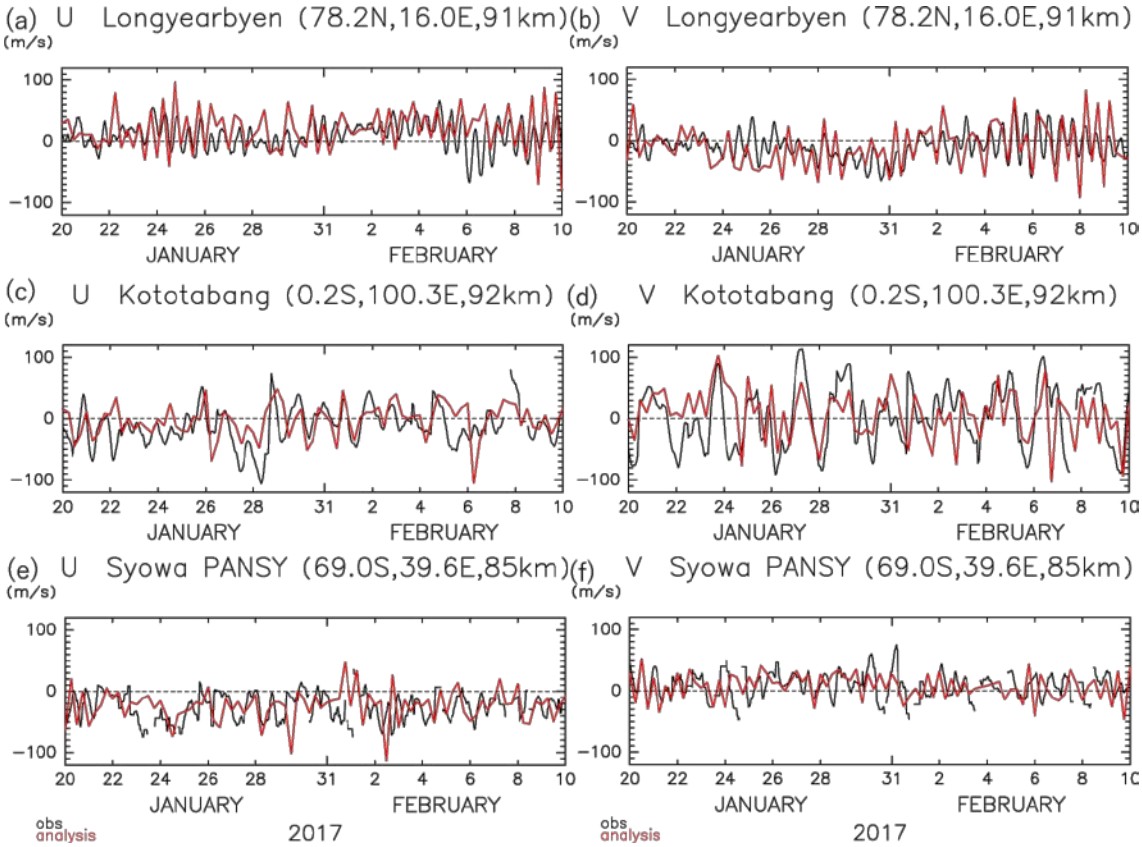

**Figure 21: The time series of the zonal (left column) and meridional (right column) wind from 6-hourly analysis (red curves) and from observation (black curves) (a) and (b) by a meteor radar at Longyearbyen in the Arctic at a height of 91 km, (c) and (d) by a meteor radar at Kototabang near the equator at 92km, and (e) and (f) by the PANSY radar at Syowa Station in the Antarctic at 85 km. Although the time intervals of the radar observation data are 1 h for meteor radars and 30 min for the PANSY radar, the 6-h running mean time series are plotted.**
