# Peer review of "An ensemble Kalman filter data assimilation system for the whole neutral atmosphere"

_Geoscientific Model Development, 2019_

## Referee Comment (RC1) · Anonymous Referee #1 · 5 Dec 2019

The paper presents the development of an ensemble data assimilation system that extends into the lower thermosphere. The background model is the Japanese Atmospherics General Circulation model for Upper Atmosphere Research (JAGUAR), and the data assimilation is provided by a 4D-LETKF. As there are few existing data assimilation models that extend into the lower thermosphere, the newly developed JAGUAR data assimilation system is beneficial to the middle atmosphere community. The paper is well written, with a clear explanation of the model, as well as examples demonstrating the performance of the data assimilation system. I recommend the paper for publication following some revisions. Specific comments are provided below.

1. Aura MLS measures temperatures, yet the results in Figures 2 and 4 use MLS observations of winds to assist in evaluating the specification of horizontal diffusion

and gravity wave drag. Presumably the winds are gradient winds based on the MLS temperature/geopotential height observations. If this is correct, it should be clearly stated in the text that the results are based on gradient winds.

2. I have some concerns about the bias correction that is applied to the MLS temperature observations. The bias correction is determined based on comparing MLS and SABER temperature observations, and then correcting the MLS temperatures. Previous studies (e.g., Hoppel et al., 2008, doi:10.5194/acp-8-6103-2008; Pedatella et al., 2014, doi:10.1002/2014JD021776) took the exact opposite approach, correcting the SABER temperatures to the MLS observations. It is unclear why the authors adjust MLS to SABER, as opposed to what has been done previously. Could the authors provide some justification for their approach?

3. An additional comment concerning the bias correction is the fact that SABER observes different local times, while MLS is fixed in local time. My understanding is that the bias correction is not restricted to only times when the two satellites are co-planar (i.e., observing similar local times). Is it possible that some of the bias correction is related to sampling different local times, and thus represents true differences that should not be removed from the observations?

4. In the discussion of the different model and data assimilation parameters, the authors often refer to the selected parameters as "optimal". Because the full parameter space is not (and cannot) be fully explored, I would recommend the authors consider revising the text since these may not be the truly "optimal" parameter settings. Rather, they represent the best among the settings that were tested.

5. The localization is performed using absolute distances. Although this approach makes sense in the horizontal, the authors may want to consider adopting a log-pressure based localization in the vertical direction. This is due to the fact that, for example, a 10 km height difference in the troposphere is much different than a 10 km height difference in the mesosphere. Well I recognize that it may not be possible to

redo all of the simulations with a different vertical localization, this is something for the authors to consider in the future.

6. The localization length scale seems to be particularly large in the vertical direction. The large vertical length scales would mean that an observation will impact nearly the whole atmosphere in the vertical direction. Is this interpretation correct?

7. The authors should double check the equation for the localization in line 520. Is a minus sign missing? That is, should this equation by be R' = R*exp[ -d**2 / (2*L**2)]?

8. OmF statistics are used in evaluating the impact of the assimilation window length. Because the error will grow during the forecast window, one would expect to have larger OmF for longer assimilation windows, which is exactly what is shown in the results. This is one reason for the OmF being smallest for a 3 h assimilation window, and largest for a 12 h assimilation window. The authors should acknowledge this aspect of evaluating the assimilation window length using OmF statistics.

9. In Figure 20, there is a horizontal line at 0.1 hPa, and it is stated in the text (lines 656-657) that "A thick horizontal bar shows the 0.1 hPa level up to which MERRA-2 pressure level data are provided". However, the MERRA-2 results shown in Figures 20e-h include results above the 0.1 hPa level. Where are these results from?

10. Line 406: "Figure 2b" should be "Figure 2d"

11. Line 434: "Figure 4b" should be "Figure 4d"

---

## Referee Comment (RC2) · Anonymous Referee #2 · 6 Jan 2020

Comments for the Author:

Described in this paper is the setup and execution of an LETKF data assimilation system for a global general circulation model with T42 horizontal resolution and 124 levels in the vertical covering from surface to about ∼150km altitude. The observations assimilated included the conventional PREBUFR data set from NCAR and the Aura Microwave Limb Sounder observations for the stratosphere and mesosphere. The setup was verified using meteor radar wind observations from 80 to 92km, that were not assimilated by the LETKF, as well as comparison with the MERRA-2 reanalysis data set covering up to 0.1 hPa (or about 60km).

Overall Comments:
**1 The paper spends a lot of time tuning and setting up the parameters for the LETKF, as makes sense for a paper like this, and you do look into the sensitivity to model diffusion as well as model gravity wave drag parameterization; however, it would be nice to see some discussion, if not some results, of the sensitivity of the MLT model parameters to the verification data sets. I'm referring to section 2.1 where you mention radiative transfer processes, ion-drag, chemical heating, dissipation heating and molecular diffusion. Several places you mention how immature the modeling of the MLT region is, it would be good to at least give the reader a since of the order of magnitude these other model parameters have on your relative to your results.**

**2 For the tuning of the model parameters of horizontal diffusion and gravity wave source intensity (figures 2 and 4) it seems you use the PANSY radar at the Syowa Station in the Antarctic and MLS observations. First, is the MLS observations used in both figures is not discussed in the text, please include statements on how the MLS observations used to make the figures, i.e. are these only MLS found at 69 degrees south and then averaged over the what period? Second, is tuning of the model parameters only using observation from around a single latitude (it seems) indicative of the best tuning of for the rest of the atmosphere? If you looked at Longyearbyen or Kotoabang meteor radar results would you come to the same conclusion? Or if you used MLS globally averaged observations would you come to the same conclusion? Third, how is the model data from the A, B and C curves averaged in latitude? Are these the values only for 69 degrees south or is it over a range of latitudes? Fourth, if you were to pull off the model data at the lat-lon of PANSY (or the lat-lon of the MLS for that particular model state) and then average those results over the time period would you get the same result as the zonal means you show in the figure? If not, how do you justify the zonal mean method comparison that you are using—or am I not understanding your figure/method.**

**3 The results in Figure 8, 9, 10, 12, 13 and 15 show OmF results. Is the "forecast" here the background forecast of 6-hours? Have you done any analysis on the results**

[Figure]

for forecasts longer than 6-hours? Maybe this is outside of the scope of the study, but can you comment on how far out your model forecast may have skill in the MLT region?

**4 The results in Figure 9, 10, 12, 13 and 15 all show bias error with respect to observations. Is there a reason you only looked at bias? Did you look at the standard deviation error for any of these results?**

Major Comments:

Lines 221-222: Here you mention thinning the observations to $\frac{1}{4}$ of the available amount, could you give more information on this, i.e. what method did you use for thinning, did you check on the sensitivity of your results to thinning more/less or can you site someone else who used this thinning procedure?

Lines 236-237: You state that you horizontally averaged the Aura MLS observations but then on line 510 you mention the sparseness of data in the upper stratosphere. If the data is so sparse region don't you want to maximize the amount of data there? If it's a computational issue then couldn't you further reduce the observations in the troposphere so that you could have more in the MLT? Did you do any sensitivity studies on including more Aura MLS observations?

Lines 383-389: How did you arrive at the B and C profiles? Did you try others and they didn't work as well?

Line 444: "Most previous studies of data assimilation did not make this bias correction." Which studies didn't, or if it's easier which studies did. Or take out the statement.

Line 499: Why only comparing at 0.1 hPa? Did you get the same results at other levels?

Lines 547-553: For Figure 11 you are showing an average of ensemble spread over the time period of 12 Jan to 20 Feb. Can you show some sort of figure that shows the ensemble spread as it is changing through the window from cycle to cycle. What I'm getting at is that often with inflation coefficients you are trying to tune them so that the

ensemble spread is too low that it is collapsing or too high that it is endlessly expanding. Could you show a figure that captures this, or is it that none of these values caused collapse or endless expansion? If this is the case then please state so.

Line 605-606: This is definitely true for the tuning of the localization since localization and ensemble size needs to be considered at the same time. I wonder if you would be able to at least do one sensitivity study looking at the optimal localization with 200 ensemble members?

Minor Comments:

Line 307: Not sure what you mean here by "at each other grid".

Line 308: What is "a setting" parameter.

Line 333-335: You have a 6-month free run from June 1st and then you pull your initial conditions over 10 days centered on January 1st. Was this same method used for the initial conditions for 30, 90 and 200 members?

Line 406: Don't you mean Figure 2d?

Line 426: Shouldn't it be: "20 February 2016"?

Line 494: Which previous studies?

Lines 527-532: You are comparing with RMS shown in Table 2. For the other parameters you compare to bias. Can you state why you switched your statistical parameter?

Line 707: Table 1?

Table 1: First "Ctrl" line "B" is bold—shouldn't it not be bold?

Table 2: Seems it would be better to plot this up and show more levels. Why in table form?

Table 3: Seems this would be better presented as a plot or series of plots. Why in table form?

Figure 3 you use "DB", figure 4 your use P0.7, figure 8 you use "G20". . .these acronyms are not defined in Table 1 or anywhere else in the manuscript. I'm not sure how to improve this for future readers but it seems that you might be able to do something like listing the acronyms in the "Ctrl" line of Table 1.

Figure 16 and 17: Why only 40N? Where the other latitudes similar?

---

## Author Response (AR1)

Reviewer #1,

The authors greatly appreciate his/her critical reading of our manuscript and constructive comments. We have revised the manuscript as much as possible following his/her comments.

Our response to each comment is in the following.

Response to comments:

>1. Aura MLS measures temperatures, yet the results in Figures 2 and 4 use MLS observations of winds to assist in evaluating the specification of horizontal diffusion and gravity wave drag. Presumably the winds are gradient winds based on the MLS temperature/geopotential height observations. If this is correct, it should be clearly stated in the text that the results are based on gradient winds.

As indicated by the reviewer, Aura MLS measures temperatures, thus zonal winds from MLS are estimated from the geopotential height assuming the gradient wind balance. Descriptions of MLS and PANSY observations have been added to the fourth paragraph of section 3.1.1.

>2. I have some concerns about the bias correction that is applied to the MLS temperature observations. The bias correction is determined based on comparing MLS and SABER temperature observations, and then correcting the MLS temperatures. Previous studies (e.g., Hoppel et al., 2008, doi:10.5194/acp-8-6103-2008; Pedatella et al., 2014, doi:10.1002/2014JD021776) took the exact opposite approach, correcting the SABER temperatures to the MLS observations. It is unclear why the authors adjust MLS to SABER, as opposed to what has been done previously. Could the authors provide some justification for their approach?

Eckermann et al. (2009, 2018) performed the bias correction for the MLS temperatures using the mean differences between MLS (v.2.2 and v.4.2) and SABER (v.2.0) temperatures from 5 to 0.002 hPa at each level. Hoppel et al., (2008) modified SABER (v.1.7) data using MLS data (v.2.2). Note the difference in the version of the SABER data. We corrected the bias for the MLS data (v.4.2). We confirmed that the bias estimated in this study is similar to

the globally-averaged mean differences between MLS temperature and other correlative data sets shown in the Data quality document (Liversey et al., 2018) at each height. Five sentences on ll.538–546 have been added.

>3. An additional comment concerning the bias correction is the fact that SABER observes different local times, while MLS is fixed in local time. My understanding is that the bias correction is not restricted to only times when the two satellites are co-planar (i.e., observing similar local times). Is it possible that some of the bias correction is related to sampling different local times, and thus represents true differences that should not be removed from the observations?

The MLS and SABER observations in 3 h intervals have a lot of missing values because observations are sparse. These missing values are not used for the bias calculation. This means that the bias was estimated using MLS and SABER data at nearly the same local time. The fourth paragraph of section 3.2 has been revised.

>4. In the discussion of the different model and data assimilation parameters, the authors often refer to the selected parameters as "optimal". Because the full parameter space is not (and cannot) be fully explored, I would recommend the authors consider revising the text since these may not be the truly "optimal" parameter settings. Rather, they represent the best among the settings that were tested.

There are six assimilation parameters to be examined. We performed assimilation runs with almost all combinations of the parameters. Several parameter settings did not work due to computational instability. We found a parameter set which provides best assimilation results in our system. This optimal parameter set is placed as the control setting (Ctrl) and the parameter dependence of the assimilation performance is examined by using the results in which one of the parameters is changed from the Ctrl set. Five sentences on ll.595–600 have been revised.

>5. The localization is performed using absolute distances. Although this approach makes

sense in the horizontal, the authors may want to consider adopting a logpressure based localization in the vertical direction. This is due to the fact that, for example, a 10 km height difference in the troposphere is much different than a 10 km height difference in the mesosphere. Well I recognize that it may not be possible to redo all of the simulations with a different vertical localization, this is something for the authors to consider in the future.

Although we did not mention, the vertical localization has already made based on a log-pressure coordinate. It is defined by the inverse of Gaussian function (eq.23), with $L=0.6$ and $d=\ln(p_{obs}/p_0)-\ln(p_{grid}/p_0)$, where $p_{obs}$, $p_{grid}$, and $p_0$ are pressures of the observation, grid point, and surface, respectively. We used the same value of $L$ for all experiments. Two sentences on ll.654–657 have been added.

>6. The localization length scale seems to be particularly large in the vertical direction. The large vertical length scales would mean that an observation will impact nearly the whole atmosphere in the vertical direction. Is this interpretation correct?

We used the vertical localization length based on the log-pressure coordinate. The word 'horizontal' has been added to the sentence on l.644 so as to make it clear.

>7. The authors should double check the equation for the localization in line 520. Is a minus sign missing? That is, should this equation by be R' = R*exp[ -d**2 / (2*L**2)]?

The description was certainly misleading. A sentence on l.647 has been revised.

>8. OmF statistics are used in evaluating the impact of the assimilation window length. Because the error will grow during the forecast window, one would expect to have larger OmF for longer assimilation windows, which is exactly what is shown in the results. This is one reason for the OmF being smallest for a 3 h assimilation window, and largest for a 12 h assimilation window. The authors should acknowledge this aspect of evaluating the assimilation window length using OmF statistics.

The third paragraph of section 3.3.4 has been revised, following the reviewer's comment.

>9. In Figure 20, there is a horizontal line at 0.1 hPa, and it is stated in the text (lines 656-657) that "A thick horizontal bar shows the 0.1 hPa level up to which MERRA-2 pressure level data are provided". However, the MERRA-2 results shown in Figures 20e-h include results above the 0.1 hPa level. Where are these results from?

The figure of the original manuscript was made using the MERRA-2 model-level data. However, we used the pressure-level data for the analysis in most parts of this paper as described in section 4.1. Figure 20 has been revised.

>10. Line 406: "Figure 2b" should be "Figure 2d"

The description has been revised following the reviewer's comment.

>11.   Line 434: "Figure 4b" should be "Figure 4d"

The description has been revised following the reviewer's comment.

Reviewer #2,

The authors greatly appreciate his/her critical reading of our manuscript and constructive comments. We have revised the manuscript as much as possible following his/her comments.
    Our response to each comment is in the following.

Response to overall comments:
>1. The paper spends a lot of time tuning and setting up the parameters for the LETKF, as makes sense for a paper like this, and you do look into the sensitivity to model diffusion as well as model gravity wave drag parameterization; however, it would be nice to see some discussion, if not some results, of the sensitivity of the MLT model parameters to the verification data sets. I'm referring to section 2.1 where you mention radiative transfer

processes, ion-drag, chemical heating, dissipation heating and molecular diffusion. Several places you mention how immature the modeling of the MLT region is, it would be good to at least give the reader a since of the order of magnitude these other model parameters have on your relative to your results.

The model performance may depend on the parameters describing the MLT processes although we used default values of the model for this study. For example, climatological concentrations of chemical species are used for the calculation of the radiative heating rate, although the $O_3$ and NO concentrations are affected by the solar activity in a short time scale. The effects of ion-drag are neglected because it is important mainly above the height of ~200 km. The chemical heating caused by the recombination of the atomic oxygen is incorporated using a global mean vertical profile of its density, and we neglected spatial and temporal changes. The last part of section 3.1 has been added.

>2. For the tuning of the model parameters of horizontal diffusion and gravity wave source intensity (figures 2 and 4) it seems you use the PANSY radar at the Syowa Station in the Antarctic and MLS observations. First, is the MLS observations used in both figures is not discussed in the text, please include statements on how the MLS observations used to make the figures, i.e. are these only MLS found at 69 degrees south and then averaged over the what period? Second, is tuning of the model parameters only using observation from around a single latitude (it seems) indicative of the best tuning of for the rest of the atmosphere? If you looked at Longyearbyen or Kotoabang meteor radar results would you come to the same conclusion? Or if you used MLS globally averaged observations would you come to the same conclusion? Third, how is the model data from the A, B and C curves averaged in latitude? Are these the values only for 69 degrees south or is it over a range of latitudes? Fourth, if you were to pull off the model data at the lat-lon of PANSY (or the lat-lon of the MLS for that particular model state) and then average those results over the time period would you get the same result as the zonal means you show in the figure? If not, how do you justify the zonal mean method comparison that you are using—or am I not understanding your ̆ figure/method.

First comment: Zonal mean zonal wind profile calculated from the geopotential height of the MLS observation assuming the gradient wind balance for 67.5°S to 72.5 °S for the time period of 12 January to 20 February is shown.

Second comment: In the northern latitude winter, the interannual variation such as the SSW is large. In the equatorial region, the interannual variation such as the QBO is large. In contrast, it is expected from that interannual and longitudinal variations in the southern hemisphere in summer are relatively small because the Carney and Drazin's theorem indicates that planetary waves from the troposphere cannot propagate in the westward background wind in the middle atmosphere. This is the reason why we compared the observation and model only for the southern hemisphere. This point is already described on ll. 431–432.

Third comment: We used only data at a latitude. However, we confirmed that similar results are obtained if we take a slightly different latitude and/or a slightly wide latitude range.

Fourth comment: We compared zonal mean fields from free runs with those from observations.

The fourth paragraph of section 3.1.1 has been revised.

>3. The results in Figure 8, 9, 10, 12, 13 and 15 show OmF results. Is the "forecast" here the background forecast of 6-hours? Have you done any analysis on the results for forecasts longer than 6-hours? Maybe this is outside of the scope of the study, but can you comment on how far out your model forecast may have skill in the MLT region?

The OmF for the W3 (W12) is calculated using the forecast for 3 (12) h, while for the other experiments, whose assimilation window is 6 h, the forecast for 6 h is used. A sentence on ll.716–717 has been added.

Predictability of the GCM will also be studied in the near future. A sentence has been added to the last part of the concluding remarks.

>4. The results in Figure 9, 10, 12, 13 and 15 all show bias error with respect to observations. Is there a reason you only looked at bias? Did you look at the standard deviation error for any of these results?

The bias of the OmA is smaller than the standard deviation as shown in Figure 8, as an

example. A sentence on l.629–630 has been added.

Response to major comments:

>Lines 221-222: Here you mention thinning the observations to 1 4 of the available amount, could you give more information on this, i.e. what method did you use for thinning, did you check on the sensitivity of your results to thinning more/less or can you site someone else who used this thinning procedure?

The thinning procedure is the same as the ALERA2 (Enomoto et al., 2013). The second paragraph of section 2.2.1 has been revised.

>Lines 236-237: You state that you horizontally averaged the Aura MLS observations but then on line 510 you mention the sparseness of data in the upper stratosphere. If the data is so sparse region don't you want to maximize the amount of data there? If it's a computational issue then couldn't you further reduce the observations in the troposphere so that you could have more in the MLT? Did you do any sensitivity studies on including more Aura MLS observations?

While the horizontal intervals of the Aura MLS observation data along the track are much finer than the current model resolution, the cross-track intervals are much coarser (approximately 30 degrees) than the model resolution. The average we made is for the along-track direction. Moreover, this average is effective to remove gravity waves that cannot be resolved by the model. We have confirmed the importance of the averaging by comparing the results with and without the averaging (not shown). The second paragraph of section 2.2.2 has been revised.

>Lines 383-389: How did you arrive at the B and C profiles? Did you try others and they didn't work as well?

We found synoptic-scale disturbances with large amplitudes which are not observed in the free runs around 0.1 hPa for the diffusion setting of A. To reduce the amplitudes of the waves,

the lowest height where the diffusion coefficient increases exponentially with increasing height is lowered (C). However, the wave activity which should be large in the winter stratosphere was strongly damped. Thus, we reduced the diffusion at higher altitudes (B). We also performed model runs with other diffusion profiles. We show results of the first (B) and second best (C) profiles as well as the default one in the manuscript. The third paragraph of section 3.1.1 has been revised.

>Line 444: "Most previous studies of data assimilation did not make this bias correction." Which studies didn't, or if it's easier which studies did. Or take out the statement.

Five sentences on ll.538–546 have been added.

>Line 499: Why only comparing at 0.1 hPa? Did you get the same results at other levels?

We have revised Figure 8 and a sentence on l.619.

>Lines 547-553: For Figure 11 you are showing an average of ensemble spread over the time period of 12 Jan to 20 Feb. Can you show some sort of figure that shows the ensemble spread as it is changing through the window from cycle to cycle. What I'm getting at is that often with inflation coefficients you are trying to tune them so that the ensemble spread is too low that it is collapsing or too high that it is endlessly expanding. Could you show a figure that captures this, or is it that none of these values caused collapse or endless expansion? If this is the case then please state so.

The global mean temperature spreads vary slightly in time and seem stable after 13 January. For an average over a short latitudinal range, both increase and decrease are observed. A new figure (Figure 12) has been added. Two sentences about the figure have also been added to the second paragraph of section 3.3.3.

>Line 605-606: This is definitely true for the tuning of the localization since localization and ensemble size needs to be considered at the same time. I wonder if you would be able to at

least do one sensitivity study looking at the optimal localization with 200 ensemble members?

As the reviewer indicated, a larger ensemble size may allow to take a larger localization length. However, we found that a shorter localization length is needed in the lower atmosphere. Thus, the optimal value should not be changed so much. Moreover, many experiments with 200 ensemble members are not practical because much computational cost is required. Two sentences on ll.753–755 have been added.

Response to minor comments:
>Line 307: Not sure what you mean here by "at each other grid".

The sentence has been revised. (l.357: "other" has been removed.)

>Line 308: What is "a setting" parameter.

The sentence on ll.359–360 has been revised.

>Line 333-335: You have a 6-month free run from June 1st and then you pull your initial conditions over 10 days centered on January 1st. Was this same method used for the initial conditions for 30, 90 and 200 members?

For the runs with 30 ensemble members, 30 initial conditions at a time interval of 6 h are used. For runs with 90 and 200 ensemble members, the time intervals for the initial conditions are taken 4 h and 2 h, respectively. Thus, they are evenly distributed over about 10 days centered on January 1st. The last paragraph of section 2.3 has been revised.

>Line 406: Don't you mean Figure 2d?

The sentence on line 495 has been revised.

>Line 426: Shouldn't it be: "20 February 2016"?

The initial condition for the free run is taken from a climatological simulation. The sentence on l.522 has been revised.

>Line 494: Which previous studies?

The reference (Miyoshi et al., 2007) on l.613 has been added.

>Lines 527-532: You are comparing with RMS shown in Table 2. For the other parameters you compare to bias. Can you state why you switched your statistical parameter?

We also show the bias (OmA) in Figure 10. To avoid unnecessary confusion, we have changed the order of description: First for Fig. 10 (ll.657–662) and second for Table 2 (ll.663–669).

>Line 707: Table 1?

The number on l.873 has been corrected.

>Table 1: First "Ctrl" line "B" is boldâ˘Tshouldn't it not be bold? ˘

The expression has been revised.

>Table 2: Seems it would be better to plot this up and show more levels. Why in table form?

It was shown using Figure 10 that the optimal localization length depends on the height. This result is also observed in another criterion, namely RMS. So, the discussion on RMS is supplementary. The third paragraph of section 3.3.2 has been revised.

>Table 3: Seems this would be better presented as a plot or series of plots. Why in table form?

It is difficult to compare the performance of the run with changing one of seven parameters with a few plots. The table has been revised. The numerals showing better performance than Ctrl were bold faced so as to grasp the table easily.

>Figure 3 you use "DB", figure 4 your use P0.7, figure 8 you use "G20". . .these acronyms are not defined in Table 1 or anywhere else in the manuscript. I'm not sure how to improve this for future readers but it seems that you might be able to do something like listing the acronyms in the "Ctrl" line of Table 1.

The sentence of the caption of Table 1 has been revised.

>Figure 16 and 17: Why only 40N? Where the other latitudes similar?

These figures (Figs. 17 and 18) are shown to explain the estimation method of the optimal number of ensemble members. The captions have been revised.

[revised manuscript text omitted]

---

## Author Response (AR2)

Referee #1,

 The authors greatly appreciate his/her critical reading of our manuscript and constructive comments. We have revised the manuscript as much as possible following his/her comments. Our response to each comment is in the following.

Response to comments:

>1. I still do not think that the authors should refer to the Ctrl parameters as "optimal". This implies that every possible combination of parameters were tested, and these values give the best results. However, only a subset of the near-infinite number of parameter settings were tested. The authors do not know if the results would be marginally improved by, for example, changing the localization length from 600 to 625 km. Because it is not possible to determine the truly optimal parameters, the Ctrl case should really be considered to be the best among what was tested. I thus still am of the opinion that the authors should not refer to these as the optimal parameters.

We have chosen to use "best" instead of "optimal", following the reviewer's comment. It means the best among what we tested. Accordingly, several sentences ll.425–428 have been revised.

>2. Lines 277-279: The 30 degree spacing is between observations on subsequent orbits. This is different then cross track observations, which would be observations along the same orbit. This statement should be revised accordingly.

Following the reviewer's comment, the sentence on l.279 has been revised.

>3. Line 539: "collection" should be "correction"

We appreciate the reviewer's indication (l.541). The correction has been made.

>4. Lines 538-545: The details about the bias corrections used in previous studies is not entirely correct. For example, Eckermann et al. (2009) state "Here we compute a similar profile over the May–June 2007 period using version 1.07 SABER data, but use it to bias-correct SABER temperatures only at altitudes below 2.7 hPa, where SABER has a 1–3 K warm bias relative to MLS and other instruments (Schwartz et al., 2008; Remsberg et al., 2008). At higher altitudes, V1.07 SABER temperatures appear to be accurate to within (Remsberg et al., 2008) whereas V2.2 MLS temperatures have a vertically structured bias near the stratopause and a systematic cold bias throughout the mesosphere (Schwartz et al., 2008). Thus, at altitudes above we bias-correct MLS temperatures using a profile based on a subjective fit to the various profiles of mean bias of MLS relative to other satellite, suborbital and analysis temperatures plotted in Fig. 26 of Schwartz et al. (2008)." Additionally, Pedatella et al. (2016, doi:10.1002/2016JA022528) indicates that a bias correction has been applied in WACCM+DART. McCormack et al. (2017) does not state explicitly whether or not a bias correction is applied, so it is unclear whether or not a bias correction was used.

Following the reviewer's comment, we have revised the paragraph (ll.541–553).

Referee #2,

 The authors greatly appreciate his/her critical reading of our manuscript and constructive comments. We have revised the manuscript as much as possible following his/her comments. Our response to each comment is in the following.

Response to comments:

>Line 9: possibly change "analysis data" to "analysis data set"

The sentence has been revised following the reviewer's comment.

>Line 24: "For" should be "for"

The sentence has been revised following the reviewer's comment.

>Line 62: possibly change wording of "simultaneously proceeding" to make more sense.

The sentence on ll.63-65 has been revised following the reviewer's comment.

>Line 473: "wide" should be "wider"

The sentence has been revised following the reviewer's comment.

>Line 475: "from that" should be "that the"

The sentence has been revised following the reviewer's comment.

[revised manuscript text omitted]